# A Control-Theoretic View of Mamba on Stability and Robustness

**Liang Cao** [1]   **Weide Liu** [2]   **Zhuo Chen** [3]   **Yan Qin** [4]

## Abstract

Selective State Space Models (SSMs) such as Mamba have emerged as efficient alternatives to Transformers, achieving linear complexity through input-dependent parameterization. However, this selectivity transforms the system from linear time-invariant (LTI) to linear parameter-varying (LPV), where individually stable matrices can produce unbounded trajectories under switching. Existing work focuses on empirical performance, leaving global stability, robustness bounds, and practical certification unresolved. This paper develops a control-theoretic framework providing the first comprehensive stability and robustness analysis for selective SSMs. We prove BIBO stability by viewing selective scans as continuous-time LTI sampling and establish two-term robustness bounds with linear growth in sequence length. For general LPV systems, we provide common quadratic Lyapunov function conditions and develop algorithms to extract certificate constants directly from trained weights. These results bridge control theory and SSM architectures, enabling formal guarantees for safety-critical deployment.

## 1. Introduction

Sequence modeling lies at the heart of modern machine learning, with applications spanning natural language processing, computer vision, speech recognition, and time-series analysis. While Transformers (Vaswani et al., 2017) have dominated this landscape, their quadratic complexity in sequence length poses fundamental scalability challenges for long-context applications. This limitation has spurred

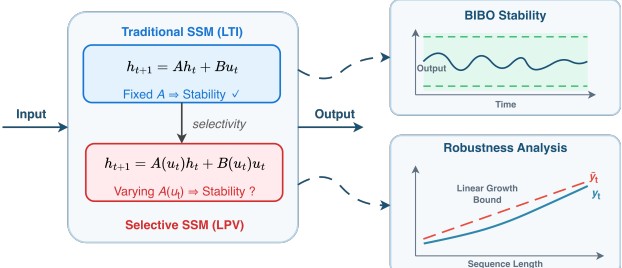

*Figure 1.* Control-theoretic analysis of selective SSMs

intense research into efficient alternatives that maintain modeling capacity while achieving sub-quadratic complexity.

State Space Models have emerged as a principled approach to this challenge, drawing on decades of control theory to enable efficient sequence processing (Gu et al., 2022). The key insight is that linear dynamical systems admit both recurrent and convolutional views, enabling hardware-efficient implementations via parallel scans (Smith et al., 2023). The HiPPO framework (Gu et al., 2020) provided theoretical grounding by connecting SSM initializations to optimal polynomial projections of signal history, yielding interpretable memory mechanisms.

A major breakthrough came with Mamba (Gu & Dao, 2024), which introduced selective state spaces where SSM parameters depend on the input content. This modification enables the model to dynamically adjust its information routing based on context, analogous to the selective attention in Transformers but with linear complexity. Mamba and its variants have achieved remarkable empirical success across modalities (Dao & Gu, 2024; Zhu et al., 2024).

However, this selectivity fundamentally changes the system's mathematical structure. Traditional SSMs are Linear Time-Invariant (LTI) systems, whose stability is determined by spectral properties of a fixed transition matrix. Selective SSMs, in contrast, are Linear Parameter-Varying (LPV) or switched systems, where the transition matrix changes at each step based on input. A critical result from control theory is that a collection of individually stable matrices can produce unstable behavior under arbitrary switching (Liberzon, 2003). This raises fundamental questions: Under what conditions are selective SSMs globally stable? How robust are outputs to input perturbations? Can we provide

[1]Department of Chemical and Biological Engineering, University of British Columbia, Vancouver, BC, Canada [2]School of Computing and Artificial Intelligence, Jiangxi University of Finance and Economics, Nanchang, China [3]Peng Cheng Laboratory, Shenzhen, China [4]Chongqing University, Chongqing, China. Correspondence to: Yan Qin <yan.qin@cqu.edu.cn>.

*Proceedings of the 43rd International Conference on Machine Learning*, Seoul, South Korea. PMLR 306, 2026. Copyright 2026 by the author(s).

verifiable certificates from network weights?

This paper develops a control-theoretic framework that, for the first time, provides verifiable stability and robustness guarantees for selective SSMs (as shown in Figure 1). Our key insight is that the fixed-$A$ exponential scan $A_t = e^{\Delta_t A}$ can be viewed as exact sampling of a continuous-time LTI system, which transforms a challenging switched-system analysis into tractable continuous-time bounds. This perspective yields three main results:

- **BIBO Stability**: The standard Mamba parameterization is provably stable under mild assumptions on the Hurwitz matrix $A$.

- **Linear-Growth Robustness**: Output sensitivity decomposes into forcing and time-warp terms, growing only *linearly* in sequence length, not exponentially as in generic switched systems.

- **Computable Certificates**: All stability and robustness constants can be extracted directly from trained weights, enabling deployment-time verification.

We further extend to general input-dependent $A(u_t)$ via common quadratic Lyapunov functions with efficient Linear Matrix Inequality (LMI) verification. Together, these contributions establish a principled bridge between classical control theory and modern selective SSM architectures, providing practitioners with actionable tools for certifying model behavior prior to deployment.

## 2. Related Work

### 2.1. State Space Models for Sequence Modeling

The application of state space models to deep learning sequence modeling has a rich recent history. HiPPO (Gu et al., 2020) introduced the idea of using structured SSM initializations derived from optimal polynomial projections, providing interpretable long-range memory. S4 (Gu et al., 2022) achieved breakthrough performance on the Long Range Arena benchmark by exploiting diagonal-plus-low-rank structure for efficient computation. S5 (Smith et al., 2023) simplified the architecture to single MIMO SSMs with parallel scan implementations. DSS (Gupta et al., 2022) and related work explored diagonal SSM parameterizations.

The gap between SSMs and Transformers for language modeling was addressed by H3 (Fu et al., 2023), which identified key capabilities (recall, comparison) where SSMs underperformed and proposed hybrid architectures. Hyena (Poli et al., 2023) provided an alternative subquadratic operator via long convolutions with data-controlled gating.

Mamba (Gu & Dao, 2024) unified these threads by introducing input-dependent "selective" parameters while retaining

efficient parallel scan implementations. The key innovation is making $B$, $C$, and the discretization step $\Delta$ input-dependent, enabling content-based routing. Mamba-2 (Dao & Gu, 2024) further refined the architecture with structured state space duality. Extensions to vision (Zhu et al., 2024), graphs, and other modalities have followed rapidly.

### 2.2. Stability of Switched and LPV Systems

Switched linear systems, where dynamics switch among a finite set of modes, and LPV systems, where dynamics vary continuously with parameters, have been extensively studied in control theory (Liberzon, 2003; Shamma & Athans, 1991). A fundamental result is that stability of individual modes does not imply stability of the switched system; counterexamples show that stable matrices can produce unbounded trajectories under fast switching.

Common Quadratic Lyapunov Functions (CQLFs) provide a clean sufficient condition: if a single positive-definite matrix $P$ satisfies $A^\top P A \preceq \beta P$ for all modes, then the switched system is stable (Boyd et al., 1994). For polytopic systems where $A \in \text{conv}\{A_1, \ldots, A_K\}$, this reduces to checking vertex LMIs (Gahinet et al., 1996).

The connection between deep learning and control theory has been explored through neural ODEs (Chen et al., 2018), implicit layers (El Ghaoui et al., 2021), and recurrent network stability (Miller & Hardt, 2019). However, systematic analysis of selective SSMs from this perspective has been limited.

### 2.3. Robustness and Lipschitz Analysis

Lipschitz bounds for neural networks have received significant attention for understanding generalization (Bartlett et al., 2017) and adversarial robustness (Szegedy et al., 2014). For feedforward networks, spectral norm products provide upper bounds; for recurrent networks, the analysis is more delicate due to temporal feedback (Zhang et al., 2018).

Incremental stability and input-to-state stability (ISS) (Sontag, 2008) provide frameworks for analyzing robustness of dynamical systems. Exponential integrators (Hochbruck & Ostermann, 2010) connect to our variable-step exponential scan interpretation. Learning and identification of stable linear systems (Hardt et al., 2018) provides complementary perspectives. Our work adapts these tools to the selective SSM setting, providing explicit robustness certificates with computable constants.

## 3. Main Results

### 3.1. Problem Setup

We consider the general selective SSM recursion:

$$h_{t+1} = A(u_t)h_t + B(u_t)u_t, \tag{1}$$
$$y_t = C(u_t)h_t + D(u_t)u_t, \tag{2}$$

where $u_t \in \mathbb{R}^m$ is the input at step $t$, $h_t \in \mathbb{R}^n$ is the hidden state, $y_t \in \mathbb{R}^p$ is the output, and $A(\cdot), B(\cdot), C(\cdot), D(\cdot)$ are input-dependent parameter maps.

The most common selective SSM parameterization, used in Mamba (Gu & Dao, 2024), employs:

$$A_t = e^{\Delta_t A}, \quad \Delta_t = \Delta(u_t) \geq 0, \tag{3}$$

where $A \in \mathbb{R}^{n \times n}$ is a fixed matrix and $\Delta_t$ is an input-dependent step size. With exact zero-order hold (ZOH) discretization:

$$h_{t+1} = e^{\Delta_t A}h_t + \int_0^{\Delta_t} e^{(\Delta_t - s)A}ds \cdot B(u_t)u_t. \tag{4}$$

Define accumulated time $\tau_0 = 0$ and $\tau_{t+1} = \tau_t + \Delta_t$. We use $\|\cdot\|$ for a vector norm and its induced operator norm; for symmetric matrices, $X \preceq Y$ denotes Löwner order. Table 3 in the appendix summarizes our notation.

The key insight enabling our analysis is that the discrete selective scan exactly samples a continuous-time forced LTI system.

**Lemma 3.1** (Scan-as-sampling Identity). *Define piecewise-constant forcing* $w(\sigma) := B(u_t)u_t$ *for* $\sigma \in [\tau_t, \tau_{t+1})$. *Let* $h(\sigma)$ *solve the continuous-time system*

$$\dot{h}(\sigma) = Ah(\sigma) + w(\sigma), \quad h(0) = h_0. \tag{5}$$

*Then the discrete recursion* (4) *produces* $h_t = h(\tau_t)$ *for all* $t$.

*Proof.* Over $\sigma \in [\tau_t, \tau_{t+1})$, forcing is constant. Variation of constants gives $h(\tau_{t+1}) = e^{\Delta_t A}h(\tau_t) + \int_0^{\Delta_t} e^{(\Delta_t - s)A}ds \cdot w(\tau_t)$, matching (4). $\square$

This lemma transforms the analysis of a discrete switched system into analysis of a continuous-time LTI system with piecewise-constant forcing, and it is a much more tractable problem.

### 3.2. Assumptions

**Assumption 3.2** (Fixed-$A$ Scan with Bounded/Lipschitz Maps). Fix an input class $\mathcal{U} = \{u : \|u\| \leq \bar{u}\}$. Assume:

(A1) **Stable continuous-time core:** $A$ is Hurwitz and $\|e^{tA}\| \leq Me^{-\alpha t}$ for all $t \geq 0$.

(A2) **Bounded/Lipschitz injection:** $\sup_{u \in \mathcal{U}} \|B(u)\| \leq \bar{B}$ and $\|B(u) - B(\tilde{u})\| \leq L_B \|u - \tilde{u}\|$.

(A3) **Bounded/Lipschitz readout:** Similarly for $C, D$ with constants $\bar{C}, \bar{D}, L_C, L_D$.

(A4) **Lipschitz step size:** $\Delta(\cdot)$ is $L_\Delta$-Lipschitz on $\mathcal{U}$.

Define the following derived constants (see Table 4 in the appendix):

$$\bar{w} := \bar{B}\bar{u}, \quad L_w := \bar{B} + L_B\bar{u}, \tag{6}$$

$$\bar{h} := M \|h_0\| + \frac{M}{\alpha}\bar{w}, \tag{7}$$

$$K_{\text{force}} := \bar{C}\frac{M}{\alpha}L_w + L_C\bar{h} + \bar{D} + L_D\bar{u}, \tag{8}$$

$$K_{\text{warp}} := \bar{C}(\|A\|\bar{h} + \bar{w})L_\Delta. \tag{9}$$

With these definitions in place, we are ready to state our main result. The key insight is that the fixed-$A$ exponential parameterization (3), combined with the scan-as-sampling identity (Lemma 3.1), allows us to leverage continuous-time stability theory rather than discrete switched-system analysis. This yields tight bounds where sensitivity grows only linearly in sequence length.

### 3.3. Main Theorem

**Theorem 3.3** (BIBO Stability + Output Robustness). *Under Assumption 3.2, let* $u_{0:T-1}$ *and* $\tilde{u}_{0:T-1}$ *be two input sequences in* $\mathcal{U}$, *generating trajectories via* (4) *with step sizes* $\Delta_t = \Delta(u_t)$ *and* $\tilde{\Delta}_t = \Delta(\tilde{u}_t)$. *Let* $\delta := \max_{0 \leq k < T} \|u_k - \tilde{u}_k\|$.

*(i) BIBO Stability. For all* $t \leq T$:

$$\|h_t\| \leq Me^{-\alpha\tau_t} \|h_0\| + \frac{M}{\alpha}\bar{w}, \quad \sup_{t \leq T} \|h_t\| \leq \bar{h}. \tag{10}$$

*(ii) Two-Term Output Robustness. For every* $t \leq T$:

$$\|y_t - \tilde{y}_t\| \leq (K_{\text{force}} + t \cdot K_{\text{warp}})\delta, \tag{11}$$

*and therefore* $\max_{0 \leq t \leq T} \|y_t - \tilde{y}_t\| \leq (K_{\text{force}} + T \cdot K_{\text{warp}})\delta$.

*Proof Sketch.* Part (i) follows from Lemma 3.1 and standard continuous-time BIBO bounds. For (ii), decompose the output mismatch into state mismatch and readout sensitivity. The state mismatch further decomposes into forcing mismatch (same-time perturbation) and time-warp mismatch (timing shift). See Section B for complete proof. $\square$

*Remark* 3.4 (Linear vs. Exponential Growth). The bound (11) grows only linearly in $t$ through the $t \cdot K_{\text{warp}}$ term. This contrasts with the simple boundary of switching systems, which usually grows exponentially. The linear growth is a direct consequence of the fixed-$A$ structure and the continuous-time interpretation.

# 4. General Stability Guarantees and Practical Verification

The fixed-$A$ analysis covers the most common Mamba parameterization. We now extend to fully input-dependent transitions $A(u_t)$ and develop methods to compute all certificate constants directly from network weights.

## 4.1. Quadratic Stability for General LPV Systems

When the transition matrix itself depends on input, we require global stability certificates that hold under arbitrary switching.

**Definition 4.1** (Common Quadratic Lyapunov Function). A set of matrices $\mathcal{A}$ is quadratically stable if there exist $P \succ 0$ and $\beta \in (0,1)$ such that $A^\top P A \preceq \beta P$ for all $A \in \mathcal{A}$.

**Assumption 4.2** (LPV System with Lipschitz Schedules). On $\mathcal{U} = \{u : \|u\| \leq \bar{u}\}$, assume:

(B1) **CQLF:** $\exists P \succ 0, \beta \in (0,1)$ such that $A(u)^\top P A(u) \preceq \beta P$ for all $u \in \mathcal{U}$.

(B2) **Uniform bounds:** $\sup_{u \in \mathcal{U}} \|B(u)\| \leq \bar{B}$, $\sup_{u \in \mathcal{U}} \|C(u)\| \leq \bar{C}$, $\sup_{u \in \mathcal{U}} \|D(u)\| \leq \bar{D}$.

(B3) **Lipschitz maps:** $\|A(u) - A(\tilde{u})\| \leq L_A \|u - \tilde{u}\|$, and similarly for $B, C, D$ with constants $L_B, L_C, L_D$.

**Theorem 4.3** (CQLF $\Rightarrow$ BIBO for LPV SSMs). *Under Assumption 4.2(B1)–(B2), define $\alpha = \sqrt{\beta}$ and $M = \kappa_P := \sqrt{\lambda_{\max}(P)/\lambda_{\min}(P)}$. Then the SSM (1)–(2) is BIBO stable: for any input sequence with $\|u_t\| \leq \bar{u}$,*

$$\sup_{t \geq 0} \|h_t\| \leq M \|h_0\| + \frac{M\bar{B}}{1 - \alpha}\bar{u}. \tag{12}$$

*Proof.* The Lyapunov function $V(h) = h^\top P h$ satisfies $V(h_{t+1}) \leq \beta V(h_t) + 2\sqrt{\lambda_{\max}(P)}\bar{B}\bar{u}\sqrt{V(h_t)} + \lambda_{\max}(P)\bar{B}^2\bar{u}^2$. Converting to the $P$-weighted norm and summing the resulting geometric series yields the bound. See Section C for details. Adding the Lipschitz assumption (B3) yields an output robustness certificate analogous to Theorem 3.3. $\square$

**Corollary 4.4** (LPV Output Robustness). *Under Assumption 4.2, let $\delta = \max_k \|u_k - \tilde{u}_k\|$ and define*

$$\bar{h} := M \|h_0\| + \frac{M\bar{B}}{1 - \alpha}\bar{u}, \tag{13}$$

$$K_h := \frac{M}{1 - \alpha}\Big(L_A\bar{h} + L_B\bar{u} + \bar{B}\Big), \tag{14}$$

$$K_y := \bar{C}K_h + L_C\bar{h} + L_D\bar{u} + \bar{D}. \tag{15}$$

*Then $\max_{t \leq T} \|y_t - \tilde{y}_t\| \leq K_y\delta$, with no dependence on horizon $T$.*

---

**Algorithm 1** Polytopic CQLF Certification

**Input:** Vertices $\{A_i\}_{i=1}^K$, tolerance $\epsilon$
$\beta_{\text{lo}} \leftarrow 0, \beta_{\text{hi}} \leftarrow 1$
**while** $\beta_{\text{hi}} - \beta_{\text{lo}} > \epsilon$ **do**
    $\beta \leftarrow (\beta_{\text{lo}} + \beta_{\text{hi}})/2$
    Solve SDP: find $P \succ 0$ s.t. $A_i^\top P A_i \preceq \beta P, \forall i$
    **if** feasible **then**
        $\beta_{\text{hi}} \leftarrow \beta$; store $P$
    **else**
        $\beta_{\text{lo}} \leftarrow \beta$
    **end if**
**end while**
**Output:** $(P, \beta_{\text{hi}})$ if feasible; INFEASIBLE otherwise

---

## 4.2. Polytopic LMI Verification

When $A(u)$ lies in a polytope $\text{conv}\{A_1, \ldots, A_K\}$, quadratic stability reduces to a finite set of LMIs.

**Proposition 4.5** (Vertex Sufficiency). *If there exist $P \succ 0$ and $\beta \in (0,1)$ such that $A_i^\top P A_i \preceq \beta P$ for all vertices $i = 1, \ldots, K$, then $A^\top P A \preceq \beta P$ for all $A \in \text{conv}\{A_1, \ldots, A_K\}$.*

*Proof.* For $A = \sum_i \xi_i A_i$ with $\xi_i \geq 0$ and $\sum_i \xi_i = 1$, the quadratic form $A^\top P A$ is convex in $A$, yielding $A^\top P A \preceq \sum_i \xi_i A_i^\top P A_i \preceq \beta P$. $\square$

This enables efficient certification via semidefinite programming. Algorithm 1 uses bisection to find the tightest contraction rate $\beta$.

## 4.3. Computing Constants from Network Weights

We now show how to extract the certificate constants $(M, \alpha, \bar{B}, L_B, \ldots)$ from a trained selective SSM. Combined with Theorem 3.3, this yields deployment-time robustness guarantees.

For the common case of diagonal $A = -\text{diag}(\lambda_1, \ldots, \lambda_n)$ with $\lambda_i > 0$, the semigroup constants are $M = 1$ and $\alpha = \min_i \lambda_i$. For general Hurwitz $A$, solving the Lyapunov equation $A^\top P + PA = -I$ yields $M = \kappa_P$ and $\alpha = 1/(2\lambda_{\max}(P))$.

The remaining constants depend on the specific parameterization of the input-dependent maps. The following result covers the standard Mamba architecture.

**Proposition 4.6** (Lipschitz Constants for Mamba Parameterizations). *Consider the standard Mamba parameterizations: step-size map $\Delta(u) = \Delta_{\min} + \Delta_r\sigma(W_\Delta u + b_\Delta)$ where $\sigma$ is the sigmoid, and gated injection $B(u) = \text{diag}(g_B(u))W_B$ with $g_B(u) \in [0,1]^n$ and $g_B$ being $L_{g_B}$-Lipschitz. Then the*

---

**Algorithm 2** Certificate Extraction for Fixed-$A$ Scans

---

**Input:** Trained model with fixed $A$, parameter networks for $B, C, D, \Delta$; input bound $\bar{u}$; horizon $T$

Compute $(M, \alpha)$ from eigenstructure of $A$

Compute $(\bar{B}, L_B), (\bar{C}, L_C), (\bar{D}, L_D), L_\Delta$ via Proposition 4.6

Compute derived constants:

$\quad \bar{w} \leftarrow \bar{B}\bar{u}, \quad L_w \leftarrow \bar{B} + L_B\bar{u}$

$\quad \bar{h} \leftarrow M\|h_0\| + \frac{M}{\alpha}\bar{w}$

$\quad K_{\text{force}} \leftarrow \bar{C}\frac{M}{\alpha}L_w + L_C\bar{h} + \bar{D} + L_D\bar{u}$

$\quad K_{\text{warp}} \leftarrow \bar{C}(\|A\|\bar{h} + \bar{w})L_\Delta$

**Output:** Certificate $\max_{t\leq T}\|y_t - \tilde{y}_t\| \leq (K_{\text{force}} + TK_{\text{warp}})\delta$

---

*following bounds hold:*

$$L_\Delta \leq \tfrac{1}{4}\Delta_r\|W_\Delta\|, \tag{16}$$

$$\bar{B} \leq \|W_B\|, \quad L_B \leq L_{g_B}\|W_B\|_F. \tag{17}$$

*Analogous bounds hold for $C(u)$ and $D(u)$ with the same gated structure.*

*Proof.* For (16), the sigmoid satisfies $\|\sigma'\|_\infty = 1/4$, so the chain rule gives $\|\nabla_u\Delta\| \leq \frac{\Delta_{\max}-\Delta_{\min}}{4}\|W_\Delta\|$. For (17), since $\|g_B(u)\|_\infty \leq 1$, we have $\|B(u)\| \leq \|W_B\|$. The Lipschitz bound follows from $\|\text{diag}(v)W_B\| \leq \|v\|_2\|W_B\|_F$ applied to $v = g_B(u) - g_B(\tilde{u})$. See Section D for details. $\square$

Algorithm 2 summarizes the complete extraction procedure. The extracted constants are upper bounds derived from worst-case analysis; actual robustness may be considerably better. Data-dependent refinements that exploit the empirical distribution of $u_t$ are an interesting direction for future work.

## 5. Experiments

We empirically validate our theoretical framework through systematic experiments on synthetic selective SSMs. Our experiments are organized around four objectives: (1) validating the BIBO stability and two-term robustness decomposition of Theorem 3.3, (2) ablating forcing versus time-warp contributions, (3) verifying LPV certificates via CQLF, and (4) demonstrating the scan-as-sampling identity.

We implement the experiments on a selective SSM with diagonal $A = -\text{diag}(\lambda_1, \ldots, \lambda_n)$ and input-dependent step sizes $\Delta(u) = \Delta_{\min} + (\Delta_{\max} - \Delta_{\min})\sigma(W_\Delta u)$. The experimental configuration and extracted certificate constants for a model with $n = 16$ hidden states are summarized in Table 5 (Appendix).

### 5.1. Stability and Robustness Validation

Figure 2 validates Theorem 3.3 across six complementary analyses. Figure 2(a) shows that state trajectories $\|h_t\|$ remain bounded under varying inputs. The actual state norm (green) stays well within the certificate region, with the time-varying BIBO bound $Me^{-\alpha\tau_t}\|h_0\| + \frac{M}{\alpha}\bar{w}$ (orange dashed) and the uniform bound $\bar{h}$ (red dashed) providing valid upper bounds throughout the trajectory.

Figure 2(b) demonstrates that the output error $\|y_t - \tilde{y}_t\|$ under perturbation $\delta = 0.1$ remains bounded by the theoretical certificate $(K_{\text{force}} + t \cdot K_{\text{warp}})\delta$. The actual error trajectory (green) lies entirely within the certificate region (shaded), confirming the validity of our two-term bound.

Figure 2(c) plots the maximum output error $R_T(\delta) := \max_{t\leq T}\|y_t - \tilde{y}_t\|$ versus perturbation size $\delta$ for $T = 150$. The empirical measurements (blue dots) scale linearly with $\delta$, matching the theoretical bound (dashed line) with a safety margin, confirming the linear dependence predicted by (11).

Figure 2(d) visualizes the decomposition of the robustness bound into its constituent terms: the constant forcing term $K_{\text{force}} \cdot \delta$ (blue) and the time-growing warp term $t \cdot K_{\text{warp}} \cdot \delta$ (red). The actual error (green) remains below the total bound, and the decomposition reveals that forcing dominates at early times while time-warp becomes significant for longer sequences.

A central prediction of Theorem 3.3 is that the sensitivity $R_T(\delta)/\delta$ grows linearly in $T$, not exponentially. Figure 2(e) confirms this: the empirical sensitivity (blue dots) follows a linear trend matching the theoretical prediction (orange dashed), with a linear fit (green dashed) showing excellent agreement. This linear growth is a direct consequence of the fixed-$A$ structure and continuous-time interpretation.

Figure 2(f) illustrates the input-dependent step size $\Delta_t$ over time, showing values concentrated around mean 0.300 within the range $[\Delta_{\min}, \Delta_{\max}]$. The inset histogram confirms the distribution of step sizes, demonstrating that the selectivity mechanism produces meaningful variation in discretization.

To demonstrate that our certificate extraction pipeline (Algorithm 2) generalizes across architectures, we sweep three axes: state dimension $n \in \{8, 16, 32, 64\}$, stability margin $\alpha \in \{0.3, 0.5, 1.0, 2.0\}$, and step-size range $[\Delta_{\min}, \Delta_{\max}]$. For each configuration we extract the constants $\bar{B}, \bar{C}, L_\Delta$ from the network weights via Proposition 4.6, compute the certificates $\bar{h}, K_{\text{force}}, K_{\text{warp}}$ from Theorem 3.3, and compare against measured values.

Table 1 reports the results. Three trends emerge: **(i)** Increasing $n$ raises both $\bar{B}$ and $\bar{C}$ (larger weight matrices), which propagates into larger $K_{\text{force}}$ and $K_{\text{warp}}$. **(ii)** Increasing $\alpha$ tightens $\bar{h}$ (from 3.11 at $\alpha$=0.3 to 0.47 at $\alpha$=2.0) and corre-

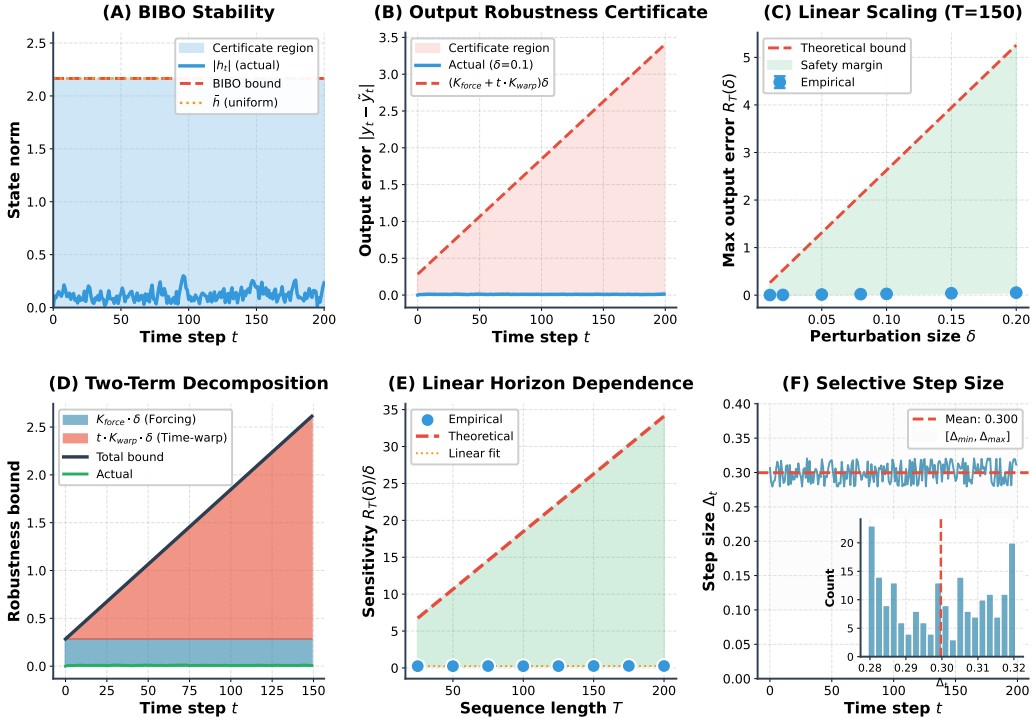

*Figure 2.* **Theorem 3.3 Validation.** (a) Bounded state trajectories. (b) Output error within certificate region. (c) Linear scaling with $\delta$. (d) $K_{\text{force}}/K_{\text{warp}}$ decomposition. (e) Linear growth in $T$. (f) Step-size distribution.

spondingly reduces $K_{\text{force}}$ ($7.80 \to 1.33$), confirming that a stronger stability margin yields tighter certificates. **(iii)** The step-size range only affects $K_{\text{warp}}$ through $L_{\Delta}$, while $K_{\text{force}}$ stays constant at $4.76$ – precisely the two-term decomposition predicted by Theorem 3.3. In all nine configurations, $\max_t \|h_t\| < \bar{h}$, validating the BIBO certificate. The robustness bound $(K_{\text{force}} + TK_{\text{warp}})\delta$ is conservative relative to the measured worst-case output error, consistent with the worst-case nature of our analysis.

### 5.2. Forcing vs. Time-Warp Ablation

Theorem 3.3 decomposes output sensitivity into a constant forcing term $K_{\text{force}}$ and a time-growing warp term $t \cdot K_{\text{warp}}$. We isolate these contributions through controlled ablations by varying the step-size range $(\Delta_{\max} - \Delta_{\min})$.

Figure 3(a) shows error trajectories $\|y_t - \tilde{y}_t\|$ under five configurations: Frozen ($\Delta_t \equiv 0.05$), Small, Medium, Large, and Extreme step-size ranges. When step size is frozen, error remains nearly constant over time; as step-size variability increases, the error growth rate increases correspondingly.

Figure 3(b) decomposes the certificate into $K_{\text{force}}$ (constant across configurations at 2.8) and $T \cdot K_{\text{warp}}$ (increasing with step-size range). This confirms that the forcing contribution is independent of step-size parameterization while time-warp scales with $L_{\Delta}$. Figure 3(c) plots $K_{\text{warp}}$ versus $L_{\Delta}$

(step-size Lipschitz constant), revealing a clear linear relationship as predicted by (9). The frozen configuration yields $K_{\text{warp}} = 0$, while extreme configurations reach $K_{\text{warp}} \approx 0.7$. Table 2 quantifies the ablation: freezing $\Delta_t \equiv 0.05$ eliminates time-warp sensitivity (slope $\approx 0$ in $T$), while amplifying $(\Delta_{\max} - \Delta_{\min})$ increases $K_{\text{warp}}$ proportionally. The measured slopes in $T$ closely match the predicted $K_{\text{warp}}$ values.

Figure 3(d) shows the step-size distributions across configurations, with the frozen case producing a delta function at $\Delta = 0.3$ and increasing spread for more sensitive configurations. Figure 3(e) isolates the time-warp effect by comparing the frozen configuration (no time-warp) against the extreme configuration (maximum time-warp), decomposing each into forcing and time-warp contributions. Figure 3(f) compares actual maximum errors at $T = 150$ against theoretical bounds. The actual errors (blue) remain well below the theoretical bounds (orange) across all configurations. The gap of 2–3 orders of magnitude reflects the worst-case nature of our analysis; tightening these bounds via data-dependent techniques is an important direction for future work.

### 5.3. LPV Certificates and Stiffness Benchmark

For architectures with fully input-dependent $A(u)$, we verify the CQLF-based certificates of Theorem 4.3. Figure 4(a) shows the eigenvalue distribution of vertex matrices

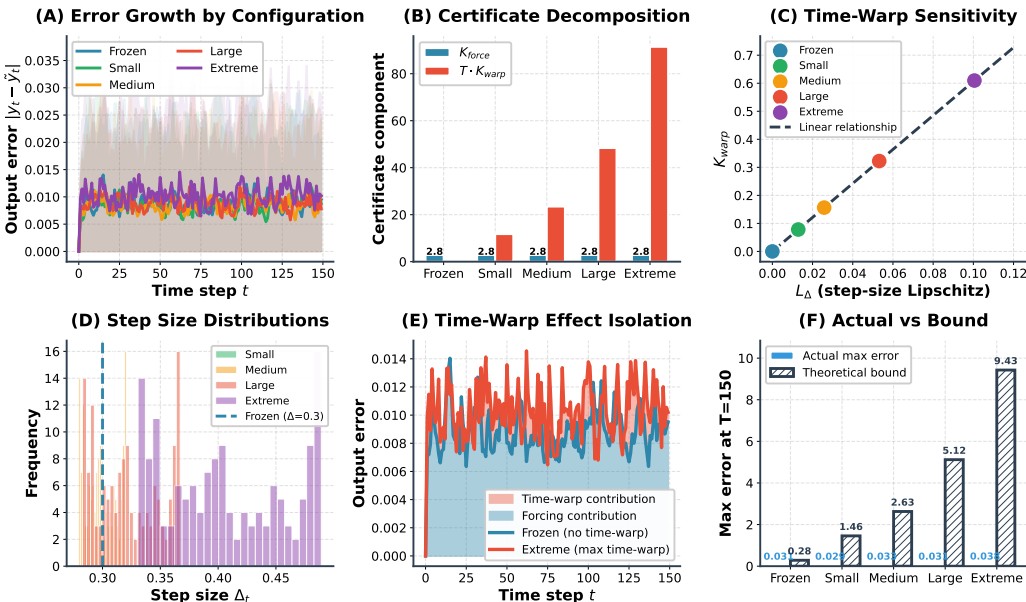

*Figure 3.* **Forcing vs. Time-Warp Ablation.** (a) Error trajectories under varying step-size ranges. (b) Forcing/warp decomposition. (c) $L_\Delta$ vs. $K_{\mathrm{warp}}$. (d) Step-size distributions. (e) Isolated time-warp effect. (f) Bound vs. actual comparison.

*Table 1.* Certificate extraction across model configurations ($T = 150$, $\delta = 0.1$, $\bar{u} = 1.0$).

| $n$ | $\alpha$ | $[\Delta_{\min}, \Delta_{\max}]$ | Extracted Constants | | | Certificates | | | Validation | | |
|---|---|---|---|---|---|---|---|---|---|---|---|
| | | | $\bar{B}$ | $\bar{C}$ | $L_\Delta$ | $\bar{h}$ | $K_{\mathrm{force}}$ | $K_{\mathrm{warp}}$ | $\max \|h_t\|$ | Bound | Actual |
| 8 | 0.5 | $[0.1, 1.0]$ | 0.68 | 0.54 | 0.195 | 1.35 | 1.76 | 0.285 | 0.26 | 4.46 | 0.017 |
| 16 | 0.5 | $[0.1, 1.0]$ | 0.93 | 0.84 | 0.195 | 1.86 | 4.76 | 0.608 | 0.31 | 9.60 | 0.021 |
| 32 | 0.5 | $[0.1, 1.0]$ | 1.25 | 0.97 | 0.195 | 2.50 | 9.05 | 0.945 | 0.50 | 15.08 | 0.018 |
| 64 | 0.5 | $[0.1, 1.0]$ | 1.81 | 1.74 | 0.195 | 3.61 | 29.10 | 2.455 | 0.69 | 39.74 | 0.028 |
| 16 | 0.3 | $[0.1, 1.0]$ | 0.93 | 0.84 | 0.195 | 3.11 | 7.80 | 0.811 | 0.37 | 12.94 | 0.022 |
| 16 | 1.0 | $[0.1, 1.0]$ | 0.93 | 0.84 | 0.195 | 0.93 | 2.47 | 0.456 | 0.23 | 7.09 | 0.021 |
| 16 | 2.0 | $[0.1, 1.0]$ | 0.93 | 0.84 | 0.195 | 0.47 | 1.33 | 0.380 | 0.17 | 5.83 | 0.020 |
| 16 | 0.5 | $[0.1, 0.5]$ | 0.93 | 0.84 | 0.087 | 1.86 | 4.76 | 0.270 | 0.24 | 4.53 | 0.020 |
| 16 | 0.5 | $[0.1, 2.0]$ | 0.93 | 0.84 | 0.412 | 1.86 | 4.76 | 1.284 | 0.40 | 19.73 | 0.022 |

*Table 2.* Ablation study: forcing vs. time-warp contributions under different step-size configurations.

| Configuration | $L_\Delta$ | $K_{\mathrm{warp}}$ | Slope in $T$ |
|---|---|---|---|
| Frozen ($\Delta_t \equiv 0.05$) | 0 | 0 | $\approx 0$ |
| Baseline ($\Delta_r = 0.09$) | 0.02 | 0.12 | 0.12 |
| Amplified ($\Delta_r = 0.18$) | 0.04 | 0.24 | 0.23 |
| High-sensitivity ($\Delta_r = 0.36$) | 0.08 | 0.48 | 0.46 |

$\{A_1, A_2, A_3, A_4\}$ in the complex plane. All eigenvalues lie within the $\sqrt{\beta}$ circle (dashed), which is strictly inside the unit circle, confirming quadratic stability of the polytopic system. Figure 4(b) visualizes the Lyapunov function $V(h) = h^\top P h$ as contours in the $(h_1, h_2)$ plane, demonstrating the quadratic structure of the certificate.

Figure 4(c) shows the distribution of empirical contraction

ratios $V(h_{t+1})/V(h_t)$ under arbitrary switching. The certified bound $\beta = 0.50$ (dashed line) is conservative relative to the observed maximum contraction ratio of $0.319$, confirming that the CQLF provides a valid certificate. We connect variable-step exponential scans to stiff dynamics modeling by training on trajectories from a stiff ODE with fast component $\lambda = 100.0$. Figure 4(d) compares tracking of the fast component: the ground truth (solid), fixed-step baseline (dashed), and variable-step method (dotted). The variable-step approach more accurately captures the rapid initial transient.

Figure 4(e) shows integration error over time on a logarithmic scale. The variable-step (adaptive) method achieves errors 2–3 orders of magnitude smaller than the fixed-step baseline ($dt = 0.01$) throughout the trajectory. Figure 4(f) illustrates the adaptive step-size selection mechanism, show-

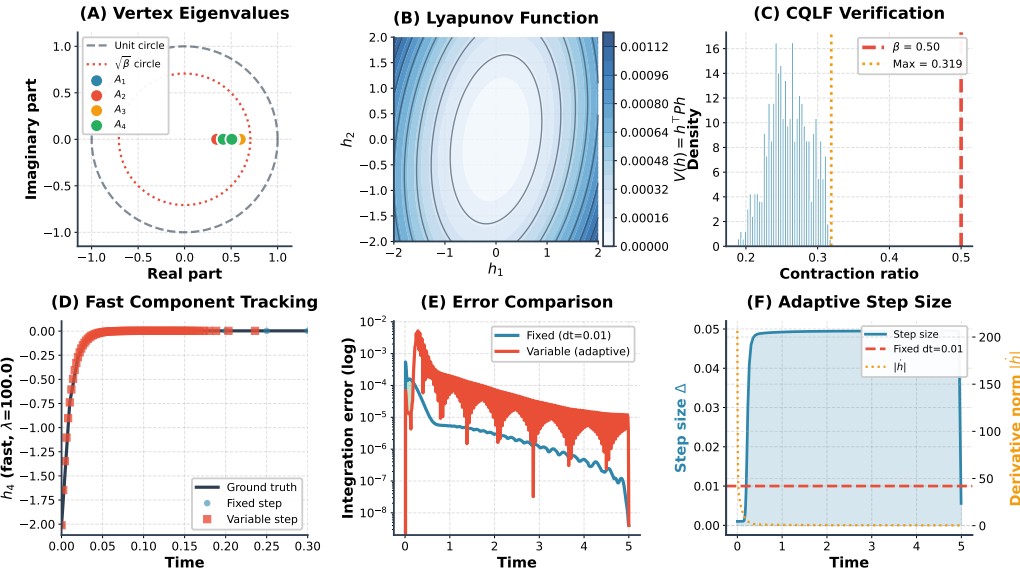

*Figure 4.* **LPV Certificate and Stiffness Benchmark.** (a) Eigenvalues within $\sqrt{\beta}$ circle. (b) Lyapunov decay under switching. (c) Contraction ratio distribution. (d) Stiff ODE tracking. (e) Variable- vs. fixed-step error. (f) Adaptive step sizes.

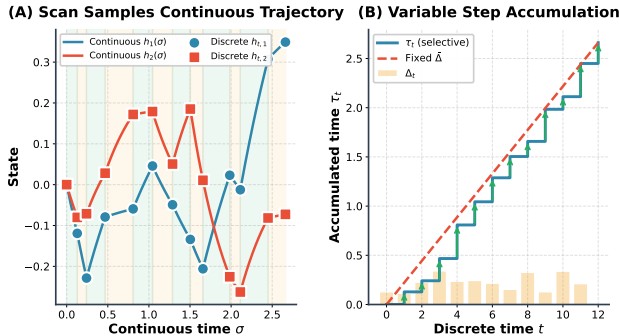

*Figure 5.* **Scan-as-Sampling Identity.** (a) Continuous trajectory vs. discrete scan samples at $\tau_t$. (b) Accumulated time $\tau_t$ vs. step $t$, illustrating the time-warp effect.

ing how step sizes (blue, left axis) correlate with derivative norm $\|\dot{h}\|$ (orange, right axis). The method automatically uses smaller steps during rapid transients and larger steps during smooth evolution.

### 5.4. Scan-as-Sampling Identity

Finally, Figure 5 illustrates the scan-as-sampling identity (Lemma 3.1), which is foundational to our analysis. Figure 5(a) overlays the continuous-time trajectory $h(\sigma)$ (solid lines for components $h_1, h_2$) with discrete scan samples $h_t$ (square and circle markers) at sampling times $\tau_t$. The exact correspondence confirms that the discrete selective scan precisely samples the continuous-time forced LTI system, validating Lemma 3.1.

Figure 5(b) plots accumulated continuous time $\tau_t$ versus

discrete step $t$. Under selective (variable) step sizes, $\tau_t$ grows nonlinearly (blue solid with markers), while fixed step sizes produce linear growth (orange dashed). The individual step sizes $\Delta_t$ (red bars) illustrate the input-dependent variation. This visualization clarifies the "time-warp" effect: perturbations to input affect not only the forcing but also the effective sampling times, leading to the $t \cdot K_{\text{warp}}$ term in our robustness bound.

## 6. Conclusion

This work addressed the problem of providing formal stability and robustness guarantees for selective State Space Models, whose input-dependent parameterization lies beyond classical LTI analysis. We showed that the fixed-$A$ exponential scan is BIBO stable and that output sensitivity decomposes into a forcing term $K_{\text{force}}$ and a time-warp term $t \cdot K_{\text{warp}}$, growing only linearly in sequence length. Our main result resolves the question of whether input-dependent discretization preserves stability, while the CQLF extension covers general LPV transitions. These certificates are computable directly from trained weights, enabling formal verification at deployment time. Our framework establishes a principled bridge between control theory and modern SSM architectures. Future work includes extending the analysis to multi-layer Mamba compositions and deriving tighter data-dependent bounds beyond worst-case analysis.

## Impact Statement

This paper presents work whose goal is to advance the field of Machine Learning, and more specifically, the theoreti-

cal understanding of selective State Space Models such as Mamba. There are many potential societal consequences of our work, none which we feel must be specifically highlighted here.

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

## A. Supplementary Tables

*Table 3.* Summary of notation.

| SYMBOL | MEANING |
|---|---|
| $u_t \in \mathbb{R}^m$ | INPUT AT STEP $t$ |
| $h_t \in \mathbb{R}^n$ | HIDDEN STATE |
| $y_t \in \mathbb{R}^p$ | OUTPUT |
| $\Delta_t = \Delta(u_t)$ | INPUT-DEPENDENT STEP SIZE |
| $\tau_t = \sum_{k=0}^{t-1} \Delta_k$ | ACCUMULATED CONTINUOUS TIME |
| $(M, \alpha)$ | SEMIGROUP CONSTANTS |
| $\bar{B}, \bar{C}, \bar{D}$ | UNIFORM BOUNDS ON MAPS |
| $L_\Delta, L_B, L_C, L_D$ | LIPSCHITZ CONSTANTS |
| $K_{\text{FORCE}}, K_{\text{WARP}}$ | CERTIFICATE CONSTANTS |
| $P \succ 0, \beta \in (0,1)$ | CQLF PARAMETERS |

*Table 4.* Certificate constants and their roles.

| Quantity | Role in Certificate |
|---|---|
| $\bar{w}, L_w$ | Bounds on effective forcing $u \mapsto B(u)u$ |
| $\bar{h}$ | Uniform state bound from BIBO stability |
| $K_{\text{force}}$ | Sensitivity from forcing mismatch and readout nonlinearity |
| $K_{\text{warp}}$ | Time-warp slope: sensitivity from step-size changes |

*Table 5.* Experimental configuration and certificate constants extracted from a selective SSM with $n = 16$ hidden states.

| Parameter | Symbol | Value |
|---|---|---|
| State dimension | $n$ | 16 |
| Input bound | $\bar{u}$ | 1.0 |
| Semigroup decay | $\alpha$ | 0.5 |
| Semigroup constant | $M$ | 1.0 |
| Step size range | $[\Delta_{\min}, \Delta_{\max}]$ | $[0.1, 1]$ |
| Forcing bound | $\bar{w}$ | 0.8 |
| Forcing Lipschitz | $L_w$ | 1.5 |
| State bound | $\bar{h}$ | 1.6 |
| Forcing certificate | $K_{\text{force}}$ | 2.4 |
| Time-warp certificate | $K_{\text{warp}}$ | 0.12 |

## B. Proof of Theorem 3.3

**Lemma B.1** (Continuous-time BIBO bound). *Assume $\left\|e^{tA}\right\| \le Me^{-\alpha t}$ and $\sup_{\sigma \ge 0} \|w(\sigma)\| \le \bar{w}$. Then the solution of $\dot{h} = Ah + w$, $h(0) = h_0$ satisfies for all $\sigma \ge 0$:*

$$\|h(\sigma)\| \le Me^{-\alpha\sigma} \|h_0\| + \frac{M}{\alpha}\bar{w}.$$

*Proof.* Variation of constants: $h(\sigma) = e^{\sigma A}h_0 + \int_0^\sigma e^{(\sigma-s)A}w(s)ds$. Bound the integral by $\bar{w} \int_0^\sigma Me^{-\alpha(\sigma-s)}ds \le \frac{M}{\alpha}\bar{w}$. □

**Lemma B.2** (Continuous-time ISS / forcing mismatch bound). *Let $h, \tilde{h}$ solve $\dot{h} = Ah + w$, $\dot{\tilde{h}} = A\tilde{h} + \tilde{w}$ with the same $A$ and $h(0) = \tilde{h}(0)$. If $\sup_\sigma \|w(\sigma) - \tilde{w}(\sigma)\| \le \delta w$, then for all $\sigma \ge 0$:*

$$\left\|h(\sigma) - \tilde{h}(\sigma)\right\| \le \frac{M}{\alpha}\delta w.$$

*Proof.* $\delta h = h - \tilde{h}$ solves $\dot{\delta h} = A\delta h + \delta w$ with $\delta h(0) = 0$, hence $\delta h(\sigma) = \int_0^\sigma e^{(\sigma-s)A}\delta w(s)ds$ and the same integral bound applies. $\square$

**Lemma B.3** (Time-shift sensitivity). *Let $h$ solve $\dot{h} = Ah + w$ and suppose $\sup_\sigma \|h(\sigma)\| \leq \bar{h}$ and $\sup_\sigma \|w(\sigma)\| \leq \bar{w}$. Then for any $\sigma, \tilde{\sigma} \geq 0$:*

$$\|h(\sigma) - h(\tilde{\sigma})\| \leq (\|A\| \bar{h} + \bar{w}) |\sigma - \tilde{\sigma}|.$$

*Proof.* $h(\sigma) - h(\tilde{\sigma}) = \int_{\tilde{\sigma}}^\sigma (Ah(s) + w(s))ds$ and bound the integrand. $\square$

**Lemma B.4** (Bilinear forcing map is Lipschitz). *Let $B : \mathcal{U} \to \mathbb{R}^{n \times m}$ satisfy $\sup_{u \in \mathcal{U}} \|B(u)\| \leq \bar{B}$ and $\|B(u) - B(\tilde{u})\| \leq L_B \|u - \tilde{u}\|$, with $\mathcal{U} = \{u : \|u\| \leq \bar{u}\}$. Then $w(u) = B(u)u$ is bounded by $\bar{w} = \bar{B}\bar{u}$ and Lipschitz with $L_w = \bar{B} + L_B\bar{u}$.*

*Proof.* Bound $\|w(u)\| \leq \bar{B}\bar{u}$. For Lipschitzness, decompose $B(u)u - B(\tilde{u})\tilde{u} = B(u)(u - \tilde{u}) + (B(u) - B(\tilde{u}))\tilde{u}$ and bound each term. $\square$

**Lemma B.5** (Output lift bound). *Assume $\sup_{u \in \mathcal{U}} \|C(u)\| \leq \bar{C}$, $\sup_{u \in \mathcal{U}} \|D(u)\| \leq \bar{D}$, and Lipschitz bounds with constants $L_C, L_D$. If $\|h(\sigma)\| \leq \bar{h}$ for all $\sigma$ and $\|u\|, \|\tilde{u}\| \leq \bar{u}$, then*

$$\|C(u)h - C(\tilde{u})\tilde{h}\| \leq \bar{C}\|h - \tilde{h}\| + L_C\bar{h}\|u - \tilde{u}\|,$$

$$\|D(u)u - D(\tilde{u})\tilde{u}\| \leq (\bar{D} + L_D\bar{u})\|u - \tilde{u}\|.$$

*Proof.* For the $C$ term, add/subtract $C(\tilde{u})h$: $C(u)h - C(\tilde{u})\tilde{h} = (C(u) - C(\tilde{u}))h + C(\tilde{u})(h - \tilde{h})$. For the $D$ term, add/subtract $D(\tilde{u})u$: $D(u)u - D(\tilde{u})\tilde{u} = (D(u) - D(\tilde{u}))u + D(\tilde{u})(u - \tilde{u})$. Apply the bounds. $\square$

**Proof of Theorem 3.3.** By Lemma 3.1, the discrete scan satisfies $h_t = h(\tau_t)$ where $h(\cdot)$ solves $\dot{h} = Ah + w$ with piecewise-constant $w(\sigma) = B(u_t)u_t$. Under Assumption A(A2), Lemma A.4 gives $\sup_\sigma \|w(\sigma)\| \leq \bar{w}$ and the Lipschitz constant $L_w$. Lemma A.1 yields (10) and thus the state bound $\bar{h}$ in (7).

For robustness, write outputs as $y_t = C(u_t)h(\tau_t) + D(u_t)u_t$ and similarly for $\tilde{y}_t$. Apply Lemma A.5 to get

$$\|y_t - \tilde{y}_t\| \leq \bar{C}\|h(\tau_t) - \tilde{h}(\tilde{\tau}_t)\| + (L_C\bar{h} + \bar{D} + L_D\bar{u})\|u_t - \tilde{u}_t\|.$$

Split the state mismatch:

$$\|h(\tau_t) - \tilde{h}(\tilde{\tau}_t)\| \leq \|h(\tau_t) - \tilde{h}(\tau_t)\| + \|\tilde{h}(\tau_t) - \tilde{h}(\tilde{\tau}_t)\|.$$

The first term is a forcing-mismatch bound: by Lemma A.2, $\|h(\tau_t) - \tilde{h}(\tau_t)\| \leq \frac{M}{\alpha} \sup_\sigma \|w(\sigma) - \tilde{w}(\sigma)\|$. By Lemma A.4 and $\delta = \|u - \tilde{u}\|_\infty$, we have $\sup_\sigma \|w - \tilde{w}\| \leq L_w\delta$.

The second term is the time-warp bound: Lemma A.3 gives $\|\tilde{h}(\tau_t) - \tilde{h}(\tilde{\tau}_t)\| \leq (\|A\|\bar{h} + \bar{w})|\tau_t - \tilde{\tau}_t|$. Assumption A(A4) implies $|\tau_t - \tilde{\tau}_t| = \left|\sum_{k=0}^{t-1}(\Delta(u_k) - \Delta(\tilde{u}_k))\right| \leq tL_\Delta\delta$.

Combine the inequalities and collect constants to obtain (11) with $K_{\text{force}}$ and $K_{\text{warp}}$ as in (8)–(9). $\square$

## C. Proofs of Theorem 4.3 and Corollary 4.4

**Lemma B.1** (CQLF implies an induced norm contraction). *Under Assumption 4.2(B1), define $\alpha = \sqrt{\beta}$ and $M = \kappa_P$. Then for any schedule $u_t \in \mathcal{U}$, the homogeneous transition satisfies $\|\Phi(t, s)\| \leq M\alpha^{t-s}$.*

*Proof.* Let $V(h) = h^\top Ph$. Then $V(h_{k+1}) \leq \beta V(h_k)$, so $V(h_t) \leq \beta^{t-s}V(h_s)$. Convert to Euclidean norm with $\lambda_{\min}(P)\|h\|^2 \leq V(h) \leq \lambda_{\max}(P)\|h\|^2$. $\square$

**Lemma B.2** (Uniform state bound under CQLF). *Under Assumption 4.2(B1)–(B2), for any input sequence with $\|u_t\| \leq \bar{u}$:*

$$\sup_{t \geq 0} \|h_t\| \leq \bar{h} := M\|h_0\| + \frac{M\bar{B}}{1 - \alpha}\bar{u}.$$

*Proof.* Unroll $h_t = \Phi(t, 0)h_0 + \sum_{k=0}^{t-1} \Phi(t, k + 1)B(u_k)u_k$. Use Lemma B.1 and $\|B(u_k)\| \leq \bar{B}$, then sum the geometric series. $\square$

**Proof of Theorem 4.3.** Lemma B.1 establishes $\|\Phi(t, s)\| \le M\alpha^{t-s}$ under assumption (B1). Lemma B.2 then yields the BIBO bound (13) by unrolling the state recursion and summing the resulting geometric series. $\square$

**Lemma B.3** (LPV state incremental bound). *Under Assumption 4.2, for two inputs $(u_t)$ and $(\tilde{u}_t)$ in $\mathcal{U}$ with $h_0 = \tilde{h}_0$ and $\delta = \|u - \tilde{u}\|_\infty$:*

$$\|h_t - \tilde{h}_t\| \le \frac{M}{1-\alpha}\Big(L_A\bar{h} + L_B\bar{u} + \bar{B}\Big)\delta.$$

*Proof.* Let $\delta h_t = h_t - \tilde{h}_t$. Subtract updates:

$$\delta h_{t+1} = A(u_t)\delta h_t + (A(u_t) - A(\tilde{u}_t))\tilde{h}_t + (B(u_t) - B(\tilde{u}_t))u_t + B(\tilde{u}_t)\delta u_t.$$

Unroll with $\delta h_0 = 0$, bound using Lemma B.1 and Lipschitz assumptions, then sum the geometric series. $\square$

**Lemma B.4** (LPV output incremental bound). *Under Theorem 4.2 and state bound $\bar{h}$: $\|y_t - \tilde{y}_t\| \le \bar{C}\|h_t - \tilde{h}_t\| + (L_C\bar{h} + L_D\bar{u} + \bar{D})\delta$.*

*Proof.* Write $y_t - \tilde{y}_t = C(u_t)h_t - C(\tilde{u}_t)\tilde{h}_t + D(u_t)u_t - D(\tilde{u}_t)\tilde{u}_t$. Add/subtract and use bounds. $\square$

**Proof of Corollary 4.4.** Lemma B.2 gives (13). Lemma B.3 yields (14). Lemma B.4 combined with (14) yields (15). $\square$

*Remark* B.5 (Practical Computation of $L_{g_B}$). In Mamba, the gating $g_B(u)$ is typically parameterized as $g_B(u) = \sigma(W_g u + b_g)$ applied elementwise. By Theorem C.1, $L_{g_B} \le \frac{1}{4}\|W_g\|$. For more complex gating networks, $L_{g_B}$ can be bounded by the product of layer-wise Lipschitz constants.

# D. Proof of Proposition 4.6

Throughout this section, $\|\cdot\|$ denotes the Euclidean norm for vectors and the induced operator ( spectral ) norm for matrices, and $\|\cdot\|_F$ denotes the Frobenius norm.

**Lemma C.1** (Sigmoid is $1/4$-Lipschitz). *Let $\sigma(z) = \frac{1}{1+e^{-z}}$. Then for all $a, b \in \mathbb{R}$,*

$$|\sigma(a) - \sigma(b)| \le \tfrac{1}{4}|a - b|.$$

*Moreover, for the elementwise extension (still denoted $\sigma$) acting on vectors $z, \tilde{z} \in \mathbb{R}^r$,*

$$\|\sigma(z) - \sigma(\tilde{z})\| \le \tfrac{1}{4}\|z - \tilde{z}\|.$$

*Proof.* For the scalar claim, $\sigma$ is differentiable and $\sigma'(z) = \sigma(z)(1 - \sigma(z)) \in (0, \frac{1}{4}]$ for all $z \in \mathbb{R}$, with maximum $\frac{1}{4}$ attained at $z = 0$. By the mean value theorem, $|\sigma(a) - \sigma(b)| \le \sup_z |\sigma'(z)| \, |a - b| \le \frac{1}{4}|a - b|$.

For the vector claim, apply the scalar bound coordinatewise:

$$\|\sigma(z) - \sigma(\tilde{z})\|^2 = \sum_{i=1}^{r} |\sigma(z_i) - \sigma(\tilde{z}_i)|^2 \le \sum_{i=1}^{r} \left(\tfrac{1}{4}|z_i - \tilde{z}_i|\right)^2 = \left(\tfrac{1}{4}\right)^2 \|z - \tilde{z}\|^2,$$

which implies $\|\sigma(z) - \sigma(\tilde{z})\| \le \frac{1}{4}\|z - \tilde{z}\|$. $\square$

**Lemma C.2** (Diagonal gating bounds). *Let $v \in \mathbb{R}^n$ and let $W$ be any real matrix with compatible dimensions. Then:*

$$\|\operatorname{diag}(v)\,W\| \le \|v\|_\infty \|W\|,$$
$$\|\operatorname{diag}(v)\,W\| \le \|v\| \|W\|_F.$$

*Similarly, if $W \in \mathbb{R}^{p \times n}$, then*

$$\|W\,\operatorname{diag}(v)\| \le \|v\|_\infty \|W\|,$$
$$\|W\,\operatorname{diag}(v)\| \le \|v\| \|W\|_F.$$

*Proof.* For the first inequality, use submultiplicativity and $\|\operatorname{diag}(v)\| = \|v\|_\infty$:

$$\|\operatorname{diag}(v)W\| \leq \|\operatorname{diag}(v)\|\,\|W\| = \|v\|_\infty\|W\|.$$

For the second inequality, use $\|X\| \leq \|X\|_F$ and compute

$$\|\operatorname{diag}(v)W\|_F^2 = \sum_{i=1}^{n} v_i^2\|W_{i,:}\|_2^2 \leq \Big(\sum_{i=1}^{n} v_i^2\Big)\Big(\sum_{i=1}^{n}\|W_{i,:}\|_2^2\Big) = \|v\|^2\|W\|_F^2,$$

so $\|\operatorname{diag}(v)W\| \leq \|\operatorname{diag}(v)W\|_F \leq \|v\|\|W\|_F$.

The right-multiplication bounds for $W\operatorname{diag}(v)$ follow by the same argument, noting that $\|W\operatorname{diag}(v)\| \leq \|W\|\,\|\operatorname{diag}(v)\|$ and $\|W\operatorname{diag}(v)\|_F^2 = \sum_{j=1}^{n} v_j^2\|W_{:,j}\|_2^2 \leq \|v\|^2\|W\|_F^2$. $\qquad\square$

With these lemmas in hand, we now prove the two claimed bounds separately.

**Step-size map.** Let

$$\Delta(u) = \Delta_{\min} + (\Delta_{\max} - \Delta_{\min})\,\sigma(W_\Delta u + b_\Delta),$$

where $\sigma$ is applied elementwise (the scalar case is a special case). For any $u, \tilde{u}$,

$$\begin{aligned}
\|\Delta(u) - \Delta(\tilde{u})\| &= (\Delta_{\max} - \Delta_{\min})\,\big\|\sigma(W_\Delta u + b_\Delta) - \sigma(W_\Delta \tilde{u} + b_\Delta)\big\| \\
&\leq (\Delta_{\max} - \Delta_{\min})\cdot\tfrac{1}{4}\|W_\Delta(u - \tilde{u})\| \qquad \text{(by Lemma C.1)} \\
&\leq \tfrac{1}{4}(\Delta_{\max} - \Delta_{\min})\,\|W_\Delta\|\,\|u - \tilde{u}\|.
\end{aligned}$$

Hence the Lipschitz constant satisfies

$$L_\Delta \leq \tfrac{1}{4}(\Delta_{\max} - \Delta_{\min})\,\|W_\Delta\|.$$

**Gated injection map.** Let $B(u) = \operatorname{diag}(g_B(u))W_B$ with $g_B(u) \in [0,1]^n$ for all $u$. Since $\|g_B(u)\|_\infty \leq 1$, Lemma C.2 gives

$$\|B(u)\| = \|\operatorname{diag}(g_B(u))W_B\| \leq \|g_B(u)\|_\infty\|W_B\| \leq \|W_B\|,$$

so $\bar{B} \leq \|W_B\|$.

For the Lipschitz bound, for any $u, \tilde{u}$,

$$B(u) - B(\tilde{u}) = \big(\operatorname{diag}(g_B(u)) - \operatorname{diag}(g_B(\tilde{u}))\big)W_B = \operatorname{diag}\big(g_B(u) - g_B(\tilde{u})\big)W_B.$$

Applying Lemma C.2 (the Frobenius-based bound) yields

$$\|B(u) - B(\tilde{u})\| \leq \|g_B(u) - g_B(\tilde{u})\|\,\|W_B\|_F \leq L_{g_B}\|u - \tilde{u}\|\,\|W_B\|_F,$$

so

$$L_B \leq L_{g_B}\|W_B\|_F.$$

**Extension to $C(u)$ and $D(u)$.** If $C(u)$ (or $D(u)$) admits the same gated form, e.g. $C(u) = \operatorname{diag}(g_C(u))W_C$ or $C(u) = W_C\operatorname{diag}(g_C(u))$ with $g_C(u) \in [0,1]^r$ and $\|g_C(u) - g_C(\tilde{u})\| \leq L_{g_C}\|u - \tilde{u}\|$, then the identical argument using Lemma C.2 gives

$$\bar{C} \leq \|W_C\|, \qquad L_C \leq L_{g_C}\|W_C\|_F,$$

and likewise for $D$ with $(W_D, g_D, L_{g_D})$. This proves all claims.

