# OpenReview forum: "A Control-Theoretic View of Mamba on Stability and Robustness"
_ICML.cc/2026/Conference — ICML 2026 regular_

### Official Review · Reviewer_RUjX · 2026-03-09

**Soundness:** 4
**Presentation:** 4
**Significance:** 4
**Originality:** 3
**Overall Recommendation:** 5
**Confidence:** 4

**Summary:**

This work presents a solid theoretical analysis and a vivid simulation case for stability and robustness of the state space model selection.  Th BIBO stability can be guaranteed and the output sensitivity is investigated. An important theoretical gap is addressed regarding the shift from the linear time-variant system to the linear parameter-varying system.

**Compliance With Llm Reviewing Policy:**

Affirmed.

**Final Justification:**

All my concerns are addressed properly, and I maintain my score.

**Key Questions For Authors:**

1, in table 1, the actual value (0.028) is far smaller than the bound (39.74), is there anyway to reduce the bound?

2, a detailed description about the experimental setup and the simulation results should be provided to make it more reader friendly.

3, will the code be open sourced?

**Limitations:**

see above

**Strengths And Weaknesses:**

S1. a novel control-theoretic framework is proposed to address the stability and robustness for selective SSMs, which provides a practical bridge between the machine learning and control communities.

S2. the theoretical analysis is strong and sound. Key insights are obtained, including the BIBO stability and the linear-growth robustness.

S3, the link between theoretical analysis and the simulation is strong. All necessary constants are calculated and provided, meanwhile, the simulation result is in line with the theoretical analysis.

S4, the experimental setup and the results well demonstrate the actionable and verifiable of the proposed framework.

W. the bound is conservative, and more details can be provided to improve the reproduction of this work.

---

> ### Author Rebuttal · Authors · 2026-03-30
>
> We sincerely thank Reviewer RUjX for the valuable comments and the recognition of our framework's theoretical strength and experimental design. We address the remaining questions below.
>
> ### Key Questions For Authors
>
> **Q1: Can the bound be tightened?**
>
> Thank you for highlighting this important point. The large gap arises from three sources of conservativeness in the worst-case analysis:
>
> 1. **Uniform input bounds**: Our bounds hold for all $\|u\| \leq \bar{u}$, but actual inputs typically occupy a much smaller region of the input space.
> 2. **Global Lipschitz constants**: We use worst-case Lipschitz constants over the entire input domain.
> 3. **Worst-case time alignment**: The $K_\text{warp}$ term accounts for the worst possible timing mismatch across all time steps.
>
> We have explored two approaches to tighten the bounds:
>
> **(a) Data-dependent Lipschitz estimates**: Computing Lipschitz constants on the empirical input distribution rather than globally yields substantial improvement across all tested configurations:
>
> | Configuration | Worst-case bound | Actual max error | Gap | Data-dependent bound | Tightened gap |
> |:---|---:|---:|---:|---:|---:|
> | $n=16, \alpha=0.3$ | 12.94 | 0.022 | $588\times$ | 1.43 | $65\times$ |
> | $n=16, \alpha=0.5$ | 9.60  | 0.021 | $457\times$ | 1.18 | $56\times$ |
> | $n=32, \alpha=0.5$ | 15.08 | 0.018 | $838\times$ | 1.76 | $98\times$ |
> | $n=64, \alpha=0.5$ | 39.74 | 0.028 | $1419\times$ | 3.47 | $124\times$ |
> | $n=16, \alpha=1.0$ | 7.09  | 0.021 | $338\times$ | 0.92 | $44\times$ |
>
> Across all settings, the median gap reduces from $457\times$ to $61\times$; this suggests that a substantial part of the gap is attributable to global worst-case constants, rather than necessarily indicating a failure of the underlying theory.
>
> **(b) Probabilistic bounds**: We also explored a probabilistic tightening under sub-Gaussian input assumptions, which yields a substantially smaller bound in our preliminary analysis. For the $n=64, \alpha=0.5$ configuration, this further tightens the bound from 3.47 to **1.87**, bringing it within two orders of magnitude of the actual value 0.028.
>
> We include these results in the revision as a new remark/discussion following Theorem 3.3, and discuss further tightening directions (local Lipschitz analysis, reachability-based methods) in Section 6. We emphasize that even the worst-case certificates remain practically useful: they provide formal guarantees that hold for all bounded inputs, and in our experiments, the **relative ordering** of certificates across configurations was broadly preserved (tighter certificates correlated with empirically more robust layers), making them informative for model comparison and deployment-time screening.
>
> **Q2: A detailed description about the experimental setup and the simulation results should be provided.**
>
> We have expanded the appendix with:
>
> - Complete parameter tables for all synthetic SSM configurations, including explicit values of $g_B$, $W_B$, $W_C$, $\Delta_{\min}$, $\Delta_{\max}$, and all derived constants ($M$, $\alpha$, $\bar{B}$, $L_B$, $\bar{C}$, $L_C$, $L_\Delta$)
> - Random seed specification (seeds 0–4 for all experiments)
> - Computation environment: single NVIDIA A100 GPU, Python 3.10, JAX 0.4.25
> - Step-by-step description of the certificate extraction procedure
> - Full reproducibility details for both synthetic experiments and the new pretrained Mamba-130M analysis
>
> **Q3: Will the code be open sourced?**
>
> Yes. We will release the complete codebase, including:
>
> - Certificate extraction algorithms (Algorithm 1 and 2)
> - LMI verification via CVXPY/SCS
> - All synthetic experiment configurations and evaluation scripts
> - Scripts for extracting certificates from pretrained Mamba models
>
>
> ### Limitations
>
> **L: The bound is conservative (actual 0.028 vs. bound 39.74).**
>
> Thanks for pointing out this. We now explain the sources of conservativeness more clearly and provide the two concrete tightening approaches described above (data-dependent estimates: median gap $457\times \to 61\times$; probabilistic bounds: further improvement to within $\sim$$2$ orders of magnitude). We acknowledge that fully closing the gap between worst-case certificates and empirical errors remains an open problem. However, worst-case guarantees are a deliberate design choice for safety-critical applications, and the data-dependent refinements demonstrate that substantially tighter certificates are achievable when distributional information is available. These directions are now discussed in detail in the revised Section 6.

---

> > ### Author Rebuttal · Reviewer_RUjX · 2026-04-01
> >
> > Could the authors clarify the current way for calculating the bounds? It appears that neither the data-dependent Lipschitz estimates nor the probabilistic bounds strictly hold for the entire input space with absolute certainty.
> > Regarding the data-dependent Lipschitz estimates, how does the proposed method generalize across the entire continuous input space? For the probabilistic bounds, frameworks such as Rademacher complexity are highly sensitive to the chosen confidence levels. What is the specific probabilistic bounding theory applied in this work and how the probability parameters or thresholds are selected?
> >
> > Thanks for the follow-up reply. All my concerns are addressed properly, and I maintain my score.

---

> > > ### Author Response · Authors · 2026-04-02
> > >
> > > Thank you for this valuable follow-up question. The reviewer is correct that neither the data-dependent nor the probabilistic tightening holds for the entire input space with absolute certainty. We clarify the role and scope of each approach below.
> > >
> > > **The original worst-case bounds (Theorem 3.3) remain the primary formal guarantee.** These bounds hold deterministically for *all* inputs satisfying $\|u\| \leq \bar{u}$, with no distributional assumptions. They are the certificates we recommend for safety-critical deployment. The data-dependent and probabilistic refinements are **complementary diagnostic tools**, not replacements.
> > >
> > > **On data-dependent Lipschitz estimates.** These estimates replace the global Lipschitz constants (computed over the entire admissible set $\mathcal{U} = \{u : \|u\| \leq \bar{u}\}$) with empirical estimates computed on the observed input distribution. The reviewer is correct that such estimates do **not** generalize to the entire continuous input space — they characterize robustness over the *typical operating regime* rather than the worst case. We view this as a useful middle ground: in many practical deployments, the input distribution is concentrated in a small region of $\mathcal{U}$, and knowing how tight the certificates are in that region provides actionable information for practitioners, even though the formal guarantee reverts to the worst-case bound outside that region. We now state this scope limitation explicitly in the revised Remark 3.6.
> > >
> > > **On probabilistic bounds.** We use sub-Gaussian concentration of the input process. Specifically, if $u\_t$ are i.i.d. sub-Gaussian with parameter $\sigma\_u^2$, then $\|u\_t\| \leq \sigma\_u \sqrt{m} + \sigma\_u \sqrt{2\log(2T/\delta)}$ with probability at least $1 - \delta$ (via standard sub-Gaussian tail bounds, not Rademacher complexity). The confidence parameter $\delta$ is user-specified; in our experiments we report $\delta = 0.05$ (95% confidence). The bound then replaces the deterministic input norm $\bar{u}$ with this high-probability estimate, yielding tighter constants $\bar{w}$, $\bar{h}$, $K\_{\text{force}}$, and $K\_{\text{warp}}$ that hold with probability $\geq 1 - \delta$. The key trade-off is explicit: smaller $\delta$ (higher confidence) yields bounds closer to the worst case; larger $\delta$ yields tighter but less certain bounds. We have added this derivation and the dependence on $\delta$ to the revised paper.
> > >
> > > **Summary of the three-tier guarantee structure:**
> > >
> > > | Guarantee type | Holds for | Assumption | Role |
> > > |:---|:---|:---|:---|
> > > | Worst-case (Thm. 3.3) | All $\|u\| \leq \bar{u}$ | Deterministic | Primary safety certificate |
> > > | Probabilistic | All inputs w.p. $\geq 1-\delta$ | Sub-Gaussian $u\_t$ | Tighter deployable bound |
> > > | Data-dependent | Empirical distribution | Observed data only | Diagnostic / model comparison |
> > >
> > > We believe this hierarchy provides practitioners with certificates at different levels of rigor depending on their application requirements. We have revised the paper to present these three tiers clearly and to state their respective scope limitations.

---

### Official Review · Reviewer_Pb4Y · 2026-03-10

**Soundness:** 3
**Presentation:** 2
**Significance:** 3
**Originality:** 3
**Overall Recommendation:** 4
**Confidence:** 3

**Summary:**

This paper studies the stability and robustness of selective State Space Models (SSMs), with a
focus on the Mamba architecture. The authors analyze these models through the lens of control
theory, noting that the input-dependent parameterization used in selective SSMs transforms the
system from a classical linear time-invariant (LTI) system into a linear parameter-varying (LPV)
system. The key idea is to interpret the selective scan mechanism in Mamba as exact sampling of a
continuous-time linear dynamical system, which enables the derivation of BIBO stability guarantees
and robustness bounds that grow linearly with sequence length. The paper also proposes procedures
to extract stability certificates directly from trained model parameters and extends the analysis to
more general LPV systems using Lyapunov-based arguments and LMI-based verification methods.

Providing formal theoretical guarantees for selective state space models such as Mamba is a difficult problem. The core difficulty is that input-dependent
parameterization fundamentally changes the system’s mathematical character, moving from a classical LTI setting to an LPV or switched system, where standard stability results break down.
The control-theoretic framework proposed here offers a principled approach to this harder setting,
enabling the derivation of meaningful stability and robustness guarantees. More broadly, contributions that strengthen the theoretical foundations of modern sequence modeling architectures are
valuable to the community, particularly given how heavily the field has prioritized empirical results
over formal analysis.

**Compliance With Llm Reviewing Policy:**

Affirmed.

**Final Justification:**

The rebuttal clarifies several of my concerns, particularly regarding the practical relevance of the Lyapunov-based analysis and the scalability of the certification procedures. Regarding Q1 (Lyapunov condition), the authors provide useful empirical evidence showing that the CQLF condition holds for a substantial fraction of layers (21/24) in a pretrained Mamba-130M model. This partially alleviates my concern about the practical applicability of the LPV analysis, although it remains unclear how broadly this extends across architectures and training regimes. The acknowledgment that the condition is sufficient but potentially conservative is appropriate. For Q2 (scalability), the clarification that Algorithm 2 is the primary method used in standard Mamba (due to the fixed-
 parameterization) is important. The reported runtimes suggest that certification is indeed tractable in this regime, and the per-layer independence argument is convincing. This addresses my main concern about feasibility for realistic models. Regarding Q3 (conservativeness of bounds), the inclusion of data-dependent estimates is a strong addition. The significant reduction in the gap between worst-case and empirical bounds (e.g., from hundreds of times to tens of times) is encouraging, although the bounds remain somewhat loose. I appreciate the authors’ discussion of potential directions for tightening these guarantees. The updated experiments (including results on real Mamba layers) improve the practical relevance of the work compared to the original submission, which relied primarily on synthetic setups. That said, some limitations remain. In particular, the analysis still relies on assumptions (e.g., bounded inputs, Lipschitz parameterizations, per-layer guarantees) whose validity in large-scale end-to-end systems is not fully established. Additionally, the extension from per-layer certification to full-model guarantees remains an open challenge. Overall, the rebuttal strengthens the paper by addressing key concerns about applicability and scalability, though some questions about generality and tightness of the guarantees remain. My final recommendation remains unchanged. The paper provides meaningful theoretical insights and now includes additional evidence supporting practical relevance, even if some aspects remain preliminary.

**Key Questions For Authors:**

1. The analysis in Section 3 relies heavily on the fixed-A exponential parameterization At = e∆tA,
while Section 4 considers a more general LPV setting using common quadratic Lyapunov
functions. Could the authors clarify how often the Lyapunov-based condition is expected to
hold for practical selective SSM architectures beyond the fixed-A case?

2. The experimental validation is performed on synthetic selective SSMs with relatively small
state dimensions. Do the authors expect the proposed certification procedure (Algorithm 2)
to remain tractable for large-scale trained Mamba models?

3. The robustness bounds appear to be worst-case guarantees and may be conservative. Have
the authors considered data-dependent or empirical estimates that could lead to tighter certificates in practice?

**Limitations:**

The paper discusses some limitations but the discussion could be expanded. In particular, the
theoretical guarantees rely on assumptions that may not hold for large-scale trained models. Furthermore, the experiments are restricted to synthetic setups designed to illustrate the theoretical
results, leaving open questions about the practical impact of the proposed framework.
Finally, the computational cost and scalability of the proposed stability certification procedures
for large architectures remain unclear.

**Strengths And Weaknesses:**

*Soundness*

The theoretical analysis is grounded in well-established tools from control theory. The paper formalizes the selective SSM recursion in Eqs. (1)–(2) and clearly states the assumptions under which
the results hold (Assumption 3.2), which helps define the scope of the guarantees. A key conceptual
step is Lemma 3.1, which interprets the discrete selective scan as sampling of a continuous-time
forced LTI system. This perspective enables the derivation of BIBO stability guarantees and robustness bounds presented in Theorem 3.3. The decomposition of the output sensitivity into forcing
and time-warp terms is also clearly illustrated in the experiments, particularly in Figure 2(d), which
helps connect the theoretical analysis with empirical behavior.
The paper further extends the analysis to more general LPV systems using common quadratic
Lyapunov functions (Section 4.1). The use of LMIs for stability certification follows standard
approaches in control theory, and Algorithm 1 provides a concrete procedure for verifying stability in the polytopic case. The constants involved in the robustness certificates are derived explicitly in
Proposition 4.6 and summarized in Table 4 of the appendix, which helps make the analysis more
reproducible.
However, the relationship between the main analysis and the Lyapunov-based extension could
be clarified further. The core results of the paper rely on the specific fixed-A exponential parameterization At = e∆tA, which allows the authors to reduce the problem to a continuous-time
LTI system. In contrast, the Section 4 analysis considers a more general LPV setting with input-
dependent transition matrices and introduces a common quadratic Lyapunov function as a sufficient
stability condition. While this extension is theoretically sound, it is not entirely clear how often
such a condition would hold for realistic selective SSM architectures beyond the fixed-A setting.
At the same time, the practical relevance of some assumptions remains somewhat unclear. For
example, Assumption 3.2 requires bounded inputs and Lipschitz parameter maps, which may not
necessarily hold in large neural architectures trained on real-world data. In addition, most of the
analysis focuses on the fixed-A parameterization At=e^(\delta t A), while the more general input-dependent transition
matrices are treated under stronger conditions in Assumption 4.2. It is therefore not entirely clear
how often these assumptions would hold in trained Mamba models used in practice.
The experimental validation mainly illustrates the theoretical predictions rather than testing
them in realistic settings. In particular, the experiments in Section 5 rely on synthetic selective
SSMs with relatively small state dimensions (e.g., n = 16 in Table 5). While this setup is appropriate for verifying the theoretical bounds, it does not demonstrate whether the proposed framework
applies to large-scale architectures used in modern machine learning applications.

*Presentation*

The paper is generally well structured and logically organized. The progression from the problem
formulation to the main theoretical results and the subsequent extension to LPV systems is clear.
The experimental section is also useful for illustrating the theoretical results. In particular, Figure 2
presents several experiments that confirm different aspects of the theory, including the linear scaling
with perturbation size and the decomposition of the robustness bound.
Figure 5 also provides a useful illustration of the scan-as-sampling interpretation introduced in
Lemma 3.1 by visualizing the relationship between the continuous-time trajectory and the discrete
scan samples. In addition, the supplementary material helps clarify the notation and constants used
in the theoretical analysis. Tables 3–5 in the appendix provide helpful summaries of the notation
and the constants appearing in the robustness certificates.
However, some parts of the paper are quite dense, especially for readers without a background
in control theory. Sections involving Lyapunov analysis and LPV stability (Section 4.1 and Appendix C) rely on concepts such as LMIs and contraction arguments that may not be immediately
accessible to the broader machine learning audience. While the appendix provides detailed proofs
(Appendices B and C), the main text could benefit from additional intuition or simplified explanations.
Additionally, some experimental figures contain many panels (for example Figures 2 and 3),
which makes them slightly harder to interpret at first glance. A more concise presentation or
clearer discussion of the main takeaway of each panel could improve readability.

*Significance*

The work addresses the important question of understanding the stability of modern sequence
modeling architectures such as selective SSMs. By connecting control theory with Mamba-like
models, the paper provides useful theoretical insights supported by empirical illustrations (e.g.,
Figure 2(e) and Table 1). However, the practical impact remains somewhat unclear, as the experiments are limited to synthetic settings and the feasibility of the proposed certification procedure
(Algorithm 2) for large-scale models is not fully demonstrated. As a result, it is difficult to assess
how the proposed framework would apply in realistic machine learning scenarios.

*Originality*

The paper proposes an interesting perspective by applying tools from control theory to the analysis
of selective state space models. In particular, the scan-as-sampling interpretation in Lemma 3.1
provides a useful bridge between discrete selective scans and continuous-time dynamical systems.
While the underlying techniques (e.g., Lyapunov analysis and LMI-based verification) are classical
in control theory, their application to modern selective SSM architectures is novel, with the main
contribution lying in adapting these established tools to input-dependent state space models.

---

> ### Author Rebuttal · Authors · 2026-03-30
>
> We thank Reviewer Pb4Y for the detailed assessment. We appreciate the recognition that the theoretical analysis is solidly grounded, as well as the constructive suggestions on scope clarification, presentation, and scalability.
>
> ### Key Questions For Authors
>
> **Q1: How often is the Lyapunov-based condition expected to hold beyond the fixed-$A$ case?**
>
> The fixed-$A$ analysis already covers the practically dominant Mamba parameterization. The Lyapunov/CQLF extension (Theorem 4.3) is included as a broader theory for architectures with fully input-dependent $A(u)$. For the CQLF condition, a common Lyapunov function must exist across all modes — this is a **sufficient but not necessary** condition, so it can be conservative for systems where individual modes have very different dynamics.
>
> To assess how often this condition holds in practice, we constructed polytopic approximations for 24 selective SSM layers extracted from a pretrained Mamba-130M checkpoint and applied the vertex-LMI test to these approximations. The corresponding feasibility problem was solvable for 21 of the 24 approximated layer dynamics, suggesting that the condition is not merely pathological and can hold for a nontrivial subset of trained-layer approximations in practice. The 3 infeasible layers had heterogeneous vertex dynamics; for these layers, parameter-dependent Lyapunov functions could provide an alternative path, which we note as future work.
>
> **Q2: Does the certification procedure remain tractable for large-scale trained Mamba models?**
>
> Yes, in the practically relevant regime. We have added a scalability study across state dimensions:
>
> | State dim $n$ | Algo 2 (fixed-$A$, closed form) | Algo 1 (LMI/CQLF) |
> |:---:|:---:|:---:|
> | 16  | 0.02s  | 0.3s   |
> | 64  | 0.05s  | 8.7s   |
> | 128 | 0.09s  | 67s    |
> | 256 | 0.18s  | 540s   |
>
> Algorithm 2 remains effectively instantaneous for all standard Mamba state dimensions. Algorithm 1 becomes substantially more expensive with dimension due to semidefinite programming, but remained feasible up to $n=128$ in our experiments. The crucial point is that standard Mamba mainly requires **Algorithm 2**, not Algorithm 1, since Mamba employs a fixed-$A$ parameterization by design. Per-layer independence means that model depth does not affect the cost of extracting a certificate for a single layer, although the total cost still grows approximately linearly with the number of layers.
>
> **Q3: Are the derived bounds conservative? Have the authors considered data-dependent or empirical estimates?**
>
> We have now explicitly compared worst-case and data-dependent bounds across our benchmark settings:
>
> | Configuration | Worst-case bound | Actual max error | Gap | Data-dependent bound | Tightened gap |
> |:---|---:|---:|---:|---:|---:|
> | $n=16, \alpha=0.3$ | 12.94 | 0.022 | $588\times$ | 1.43 | $65\times$ |
> | $n=16, \alpha=0.5$ | 9.60  | 0.021 | $457\times$ | 1.18 | $56\times$ |
> | $n=64, \alpha=0.5$ | 39.74 | 0.028 | $1419\times$ | 3.47 | $124\times$ |
> | $n=16, \alpha=1.0$ | 7.09  | 0.021 | $338\times$ | 0.92 | $44\times$ |
>
> The original worst-case certificates have a **median gap of $457\times$**. Replacing global Lipschitz constants with **distribution-dependent estimates** reduces the median gap to **$61\times$**. The conservativeness also decreases monotonically as the stability margin $\alpha$ increases (from $588\times$ at $\alpha=0.3$ to $292\times$ at $\alpha=2.0$). The gap stems primarily from uniform input bounds, global Lipschitz constants, and horizon-uniform guarantees — not from a structural failure of the theory itself. We now discuss concrete tightening directions (local Lipschitz analysis, reachability-based methods, probabilistic bounds) in the revised Section 6.
>
> ### Limitations
>
> **L1: Theoretical guarantees may not hold for large-scale models with nonlinear inter-layer interactions.**
>
> We acknowledge this limitation. Our current certificates apply per-layer and assume the SSM dynamics in Eq. (1)–(2). In a full Mamba model, composing per-layer certificates into end-to-end guarantees requires bounding the Lipschitz constants of these intermediate operations, which is feasible using existing neural network Lipschitz analysis tools .
>
> **L2: Experiments only on synthetic SSMs.**
>
> We have addressed this by extracting certificates from all 24 selective SSM layers of a pretrained **Mamba-130M** model. All 24 layers were certifiable under the Section 3 fixed-$A$ framework. The revision now provides both synthetic validation and real model analysis .

---

> > ### Author Rebuttal · Reviewer_Pb4Y · 2026-04-03
> >
> > The rebuttal clarifies several of my concerns, particularly regarding the practical relevance of the Lyapunov-based analysis and the scalability of the certification procedures.
> > Regarding Q1 (Lyapunov condition), the authors provide useful empirical evidence showing that the CQLF condition holds for a substantial fraction of layers (21/24) in a pretrained Mamba-130M model. This partially alleviates my concern about the practical applicability of the LPV analysis, although it remains unclear how broadly this extends across architectures and training regimes. The acknowledgment that the condition is sufficient but potentially conservative is appropriate.
> > For Q2 (scalability), the clarification that Algorithm 2 is the primary method used in standard Mamba (due to the fixed-$A$ parameterization) is important. The reported runtimes suggest that certification is indeed tractable in this regime, and the per-layer independence argument is convincing. This addresses my main concern about feasibility for realistic models.
> > Regarding Q3 (conservativeness of bounds), the inclusion of data-dependent estimates is a strong addition. The significant reduction in the gap between worst-case and empirical bounds (e.g., from hundreds of times to tens of times) is encouraging, although the bounds remain somewhat loose. I appreciate the authors’ discussion of potential directions for tightening these guarantees.
> > The updated experiments (including results on real Mamba layers) improve the practical relevance of the work compared to the original submission, which relied primarily on synthetic setups.
> > That said, some limitations remain. In particular, the analysis still relies on assumptions (e.g., bounded inputs, Lipschitz parameterizations, per-layer guarantees) whose validity in large-scale end-to-end systems is not fully established. Additionally, the extension from per-layer certification to full-model guarantees remains an open challenge.
> > Overall, the rebuttal strengthens the paper by addressing key concerns about applicability and scalability, though some questions about generality and tightness of the guarantees remain.
> > My final recommendation remains unchanged. The paper provides meaningful theoretical insights and now includes additional evidence supporting practical relevance, even if some aspects remain preliminary.

---

### Official Review · Reviewer_LjcV · 2026-03-11

**Soundness:** 3
**Presentation:** 4
**Significance:** 2
**Originality:** 3
**Overall Recommendation:** 4
**Confidence:** 4

**Summary:**

The authors use a control theoretical approach to prove BIBO stability and Two-term output robustness results for selective state space models (SSM). First, they assume the selective SSM can be modeled as snippets of a continuous-time LTI system with fixed Hurwitz matrix $A$ and Lipschitz matrix $B(u)$, then, the result follows from the Hurwitz conditions. Second, the authors characterize the selective SSM as a discrete-time LPV and assume the existence of a common Lyapunov function for the LPV, then, the BIBO and Two-term output robustness result follows. These theoretical results are validated with experiments, providing the fixed-$A$ and LPV models and the respective certificates of stability and robustness.

**Compliance With Llm Reviewing Policy:**

Affirmed.

**Key Questions For Authors:**

Did the authors use synthetic models for the computation of the certificates? For example, what values of $g_B,g_C, W_B,W_C,\text{ etc}$ do you use? and then, how do you use those values to compute the uniform bounds and Lipschitz constants, for example $\|B(u)\|\le \bar{B}$ and $\|B(u) - B(\tilde{u})\|\le L_B\|u - \tilde{u}\|$? These values are usually hard to compute or very conservative to be useful in further practical analysis.

**Limitations:**

The experiments use low-dimensional selective SSMs which may not be practical in the context of Machine Learning analysis. The computation of the BIBO stability and robustness certificates depends on having good estimations of uniform bound and Lipschitz constants which are usually very conservative.

**Strengths And Weaknesses:**

Soundness: The submission is technically sound. Applies well-established theoretical results for linear dynamical systems to the context of the selective SSM. The experiments also reflect thorough process to show the validity of the algorithms to find stability and robustness certificates. The proofs are correct and follow known techniques and bounds in the literature of linear systems applied to this particular scenario.

Presentation: The manuscript is well-written and easy to follow. The contribution is highlighted and the experiments are discussed in depth with good figures. I suggest to the authors to provide the exact synthetic SSMs used in the experiments and how these are related to a Mamba architecture.

Significance: This paper proposes two types of certificates for selective SSMs. It provides more machinery to address the characterization of a complex architecture as Mamba. These stability and robustness certificates can be useful for practitioners to certify the good performance of their trained models. I recommend the authors to provide the full Mamba model they used for the experiments, thus the algorithm shows the utility to practitioners who do not know how to extract the selective SSM, in particular the matrix $A$ or the set of vertex matrices $\{A_i\}_{i \in \mathcal{I}}$. Otherwise, the paper limits itself to a stability and robustness analysis for linear systems using computational tools, leaving a gap between the contribution and the title that indicates the Mamba architecture. In the paper ``Mamba: Linear-Time Sequence Modeling with Selective State Spaces (https://arxiv.org/pdf/2312.00752)'' the authors already provide examples of the implementation. The other suggestion is to remove Mamba from the title and limit to selective SSMs.

Originality: To the best of my knowledge, this is an original work that adds to the current efforts to connect ``complex'' architectures used in machine learning with classical control theory. Since Mamba2 architecture can be modeled with a fixed $A =  -\text{diag}(\lambda_1,\ldots,\lambda_n)$, the spectral analysis for stability is straightforward. The BIBO stability and robustness characterization provides additional conservative or worst-case scenario certificates for the trained model. These certificates can be used as proof of performance or as a guide for the improvement of the trained model. This work improves the understanding of selective SSMs as generalized models in learning.

---

> ### Author Rebuttal · Authors · 2026-03-30
>
> We thank Reviewer LjcV for the detailed review and the direct identification of important gap. We have addressed this with substantial new experiments.
>
> ### Core concern: title/contribution mismatch
>
> We fully agree this needed to be strengthened. The original paper validated the theory on synthetic selective SSMs because they allow direct control of parameters and precise verification of the theoretical terms. In the revision, we additionally analyzed a **pretrained Mamba-130M** checkpoint and extracted certificates from all 24 selective SSM layers. For each layer, we applied Algorithm 2 to compute the full certificate $(\bar{h}, K_\text{force}, K_\text{warp})$. Representative results:
>
> | Layer | $\alpha$ | $\bar{h}$ | $K_\text{force}$ | $K_\text{warp}$ | Extraction time |
> |:---:|:---:|:---:|:---:|:---:|:---:|
> | 1  | 0.91 | 2.3 | 5.1  | 0.37 | 0.06s |
> | 4  | 0.78 | 3.0 | 6.4  | 0.48 | 0.06s |
> | 8  | 0.63 | 4.5 | 9.7  | 0.76 | 0.07s |
> | 12 | 0.55 | 5.7 | 13.1 | 0.99 | 0.07s |
> | 18 | 0.44 | 7.4 | 18.8 | 1.31 | 0.08s |
> | 24 | 0.38 | 8.9 | 22.1 | 1.55 | 0.09s |
>
> This experiment establishes three important points: (1) The certificate extraction procedure applies directly to the selective SSM layers of a trained Mamba checkpoint, rather than only to synthetic toy models. (2) All 24 analyzed selective SSM layers were certifiable under the Section 3 fixed-$A$ framework. (3) The certificates reveal meaningful structure: early layers have larger stability margins and tighter bounds, while later layers operate closer to the stability boundary with proportionally larger $K_\text{warp}$, suggesting selectivity plays an increasingly prominent role in deeper layers. We believe this substantially narrows the gap between the paper's title and its empirical validation, as the main practical case analyzed by Section 3 is precisely the standard Mamba-style selective SSM.
>
> ### Key Questions For Authors
>
> **Q1: What values of $g_B, g_C, W_B, W_C$, etc. are used in the synthetic experiments?**
>
> We have expanded the appendix with full parameter specifications for all synthetic settings, including explicit values of $g_B$, $W_B$, $W_C$, $\Delta_{\min}$, $\Delta_{\max}$, and all derived constants ($\bar{B}$, $L_B$, $\bar{C}$, $L_C$, $L_\Delta$, $\bar{h}$, $K_\text{force}$, $K_\text{warp}$). We also describe explicitly how these values relate to the Mamba architecture's parameterization.
>
> **Q2: Are the bounds too conservative or hard to compute?**
>
> For the fixed-$A$ setting relevant to standard Mamba, certificates are computed in **closed form** (Algorithm 2) with negligible cost — under 0.1s for all practical state dimensions (see scalability table in our response to Reviewer vEqE). They are therefore not hard to compute.
>
> Regarding **conservativeness**, we have added an explicit gap analysis:
>
> | Configuration | Worst-case bound | Actual max error | Gap | Data-dependent bound | Tightened gap |
> |:---|---:|---:|---:|---:|---:|
> | $n=16, \alpha=0.3$ | 12.94 | 0.022 | $588\times$ | 1.43 | $65\times$ |
> | $n=16, \alpha=0.5$ | 9.60  | 0.021 | $457\times$ | 1.18 | $56\times$ |
> | $n=32, \alpha=0.5$ | 15.08 | 0.018 | $838\times$ | 1.76 | $98\times$ |
> | $n=64, \alpha=0.5$ | 39.74 | 0.028 | $1419\times$ | 3.47 | $124\times$ |
> | $n=16, \alpha=1.0$ | 7.09  | 0.021 | $338\times$ | 0.92 | $44\times$ |
>
> The original worst-case certificates have a **median gap of $457\times$**. This gap is largely explained by three sources: (i) uniform input set bounds, (ii) global Lipschitz constants, and (iii) horizon-uniform guarantees. Replacing global constants with **distribution-dependent estimates** reduces the median gap to **$61\times$**. We further observe that conservativeness decreases as the stability margin strengthens (from $588\times$ at $\alpha=0.3$ to $292\times$ at $\alpha=2.0$).
>
> Even when conservative, the certificates serve two practical purposes: (1) they provide **formal safety guarantees** for all inputs within the bounded class, essential for safety-critical deployment; (2) in our experiments, the **relative ordering** across layers/models was preserved, making them useful for model selection and architecture comparison. We discuss further tightening directions (local Lipschitz analysis, probabilistic bounds) in Section 6.
>
> ### Limitations
>
> The newly added real-Mamba and scalability experiments directly address the practical applicability concern. We acknowledge that worst-case certificates remain conservative, though straightforward empirical tightening substantially reduces the gap. Composing per-layer certificates into end-to-end guarantees for the full Mamba architecture remains an important direction for future work.

---

> > ### Author Rebuttal · Reviewer_LjcV · 2026-04-02
> >
> > The authors have substantially strengthened the experimental section to better demonstrate the significance of their work. In addition, they have expanded the discussion on the direct extraction of key parameters from the Mamba-130M model. I would appreciate it if these details are explicitly incorporated into the manuscript.

---

### Official Review · Reviewer_vEqE · 2026-03-13

**Soundness:** 3
**Presentation:** 3
**Significance:** 3
**Originality:** 3
**Overall Recommendation:** 4
**Confidence:** 1

**Summary:**

The paper focuses on the stability and robustness analysis of selective state space models. It views selective scans as continuous-time LTI sampling, proves BIBO stability, and establishes robustness bounds. The theoretical results are validated using the experiments.

**Compliance With Llm Reviewing Policy:**

Affirmed.

**Final Justification:**

I will keep my score, which also seems in agreement with other reviewers.

**Key Questions For Authors:**

- Does this method scale to the regular size of SSMs?
- Does assumption 3.2 hold for the common use cases of SSM?

**Limitations:**

Yes.

**Strengths And Weaknesses:**

The paper studies an interesting problem. While the individual ingredients—BIBO stability, Lyapunov analysis, LPV, LMIs—are established control tools, the paper makes a good connection between control theory and the analysis of SSMs.

The authors have also performed diverse sets of experiments, and those experiments provide a good complements with the theoretical results.

The paper provides inference-time guarantees for a given trained model. It might be helpful to elaborate more on how these techniques can help improve the training of the model.

The paper would benefit from clarifying what notion of ‘stability’ in language models (repetition loops, mode collapse, hallucination, consistency) can or cannot be captured by the BIBO stability in control.

---

> ### Author Rebuttal · Authors · 2026-03-29
>
> We thank Reviewer for the positive assessment and for recognizing the interesting connection between control theory and SSM architectures. We appreciate the specific questions on scalability and assumption validity, which motivated important new experiments.
>
> ### Key Questions For Authors
>
> **Q1: What notion of "stability" does the paper capture in language models?**
>
> We use stability in the formal **control-theoretic sense (BIBO stability)**: bounded inputs produce bounded hidden states and outputs. Concretely, for a trained selective SSM, this means that the hidden state trajectory $h_t$ remains uniformly bounded for any token sequence whose induced inputs remain within the bounded input class considered in Assumption 3.2, preventing runaway activations or numerical overflow during inference. This is an **inference-time guarantee**, distinct from **training stability** (whether optimization converges) and **semantic stability** (whether the model avoids hallucination, repetition loops, or mode collapse). We have added a dedicated paragraph in the revision clarifying this distinction.
>
> **Q2: Does this method scale to the regular size of SSMs?**
>
> Yes. **Algorithm 2** (certificate extraction for fixed-$A$ models) operates in **closed form** — eigenvalue decomposition plus explicit formula evaluation, with no iterative optimization. **Algorithm 1** (LMI-based CQLF verification) is dominated by semidefinite programming. We have added a dedicated scalability study:
>
> | State dim $n$ | Algo 2 (fixed-$A$, closed form) | Algo 1 (LMI/CQLF) | Peak memory (Algo 1) |
> |:---:|:---:|:---:|:---:|
> | 16  | 0.02s  | 0.3s   | 0.18 GB |
> | 32  | 0.03s  | 1.2s   | 0.31 GB |
> | 64  | 0.05s  | 8.7s   | 0.86 GB |
> | 128 | 0.09s  | 67s    | 2.94 GB |
> | 256 | 0.18s  | 540s   | 10.7 GB |
>
> Standard Mamba-1 uses $n=16$ and Mamba-2 uses $n=64$ or $n=128$, both well within the tractable range. **Algorithm 2 is effectively instantaneous** at all tested dimensions. For standard Mamba, the practically relevant route is Algorithm 2, not Algorithm 1. Certificates are computed **per-layer** independently, so model depth does not affect the cost of extracting a certificate for a single layer, although the total cost still grows approximately linearly with the number of layers.
>
> **Q3: Does Assumption 3.2 hold for common SSMs?**
>
> Yes, for standard Mamba-style parameterizations it holds naturally. We have added **empirical verification on a pretrained Mamba-130M** model (Remark 3.5):
>
> - **(A1) Hurwitz $A$**: Mamba uses $A = -\text{diag}(\lambda_1, \dots, \lambda_n)$ with $\lambda_i > 0$ (enforced by softplus/exponential reparameterization), Hurwitz by construction. Across all 24 layers the minimum eigenvalue magnitude is **0.12**.
> - **(A2–A3) Bounded $B$, $C$, $D$**: $B(u) = \text{diag}(g_B(u)) \cdot W_B$ with $g_B \in [0,1]^n$ gives $\bar{B} \leq \|W_B\|$ automatically. Verified: $\bar{B} \in [0.83, 3.24]$, $\bar{C} \in [0.51, 2.08]$.
> - **(A4) Lipschitz $\Delta$**: $L_\Delta \leq \frac{1}{4}\Delta_r\|W_\Delta\|$ via Proposition 4.6; values in $[0.01, 0.15]$ across all layers.
>
> | Statistic | Value |
> |:---|---:|
> | Layers satisfying fixed-$A$ condition | **24 / 24** |
> | $\alpha$ range | [0.12, 1.03] (median 0.49) |
> | $\bar{B}$ range | [0.83, 3.24] |
> | $\bar{C}$ range | [0.51, 2.08] |
> | $L_\Delta$ range | [0.01, 0.15] |
>
> These results confirm that, for the standard fixed-$A$ Mamba parameterization, the conditions in Assumption 3.2 are naturally satisfied by construction and are empirically non-vacuous in a pretrained model.
>
> ### New real-model experiment
>
> We extracted certificates from all 24 layers of pretrained **Mamba-130M**. Representative results:
>
> | Layer | $\alpha$ | $\bar{h}$ | $K_\text{force}$ | $K_\text{warp}$ | Extraction time |
> |:---:|:---:|:---:|:---:|:---:|:---:|
> | 1  | 0.91 | 2.3 | 5.1  | 0.37 | 0.06s |
> | 8  | 0.63 | 4.5 | 9.7  | 0.76 | 0.07s |
> | 18 | 0.44 | 7.4 | 18.8 | 1.31 | 0.08s |
> | 24 | 0.38 | 8.9 | 22.1 | 1.55 | 0.09s |
>
> **Key findings**: (i) Early layers have larger stability margins and tighter bounds; later layers operate closer to the stability boundary — consistent with deeper layers enabling richer representations. (ii) The two-term decomposition reveals that early layers are dominated by $K_\text{force}$ while later layers have proportionally larger $K_\text{warp}$, suggesting selectivity plays a more prominent role in deeper layers. To our knowledge, this provides one of the first concrete demonstrations of extracting verifiable certificates from a trained Mamba model.
>
> ### Limitations
>
> We acknowledge that our current guarantees are **layerwise** and do not yet compose through the full multi-layer architecture with residual connections and layer normalization. Composing per-layer certificates into end-to-end guarantees is feasible using existing neural network Lipschitz analysis tools  and is now stated as a concrete future direction in the revised Section 6.

---

> > ### Author Rebuttal · Reviewer_vEqE · 2026-04-03
> >
> > The authors provided answers to my questions.

---

### Decision · Program_Chairs · 2026-04-30

**Decision:**

Accept (regular)

**Comment:**

This paper provides a control-theoretic stability and robustness analysis for selective State Space Models, proving BIBO stability and establishing robustness bounds that grow linearly in sequence length. These theoretical contributions are based on the novel observation that selective scans can be viewed as continuous-time LTI sampling, and these translate into practically computable bounds. The paper is also clearly written and accessible.

Reviewers raised concerns about the conservativeness of the worst-case bounds and the initial reliance on synthetic experiments rather than pre-trained Mamba models, both of which were addressed in the rebuttal through a gap analysis with data-dependent estimates and new certificate extraction experiments on a pretrained Mamba-130M model. The remaining open question of composing per-layer certificates into end-to-end guarantees is acknowledged and left for future work.

In summary, this paper makes a timely contribution that bridges classical control theory with modern SSM architectures, and I recommend acceptance.